# Polyatomic Complexes: A topologically informed learning representation for atomistic systems

## Abstract

Developing robust physics-informed representations of chemical structures that enable models to learn topological inductive biases is challenging. In this manuscript, we present a representation of atomistic systems. We begin by proving that our representation satisfies all structural, geometric, efficiency, and generality constraints. Afterward, we provide a general algorithm to encode any atomistic system. Finally, we report performance comparable to state-of-the-art methods on numerous tasks. We open-source all code and datasets. The anonymized code and data are available in the supplementary material.

## 1 Introduction

Recent advances in machine learning have enabled us to leverage representations of chemical state at different levels of chemical scale to learn meaningful patterns in chemical data. This is a particular kind of representation learning. In cheminformatics and bioinformatics numerous representations of chemical structures exist. The most popular representations are SMILES, ECFP fingerprints, SELFIES, and Graphs. The following section briefly presents existing molecular representations and their construction.

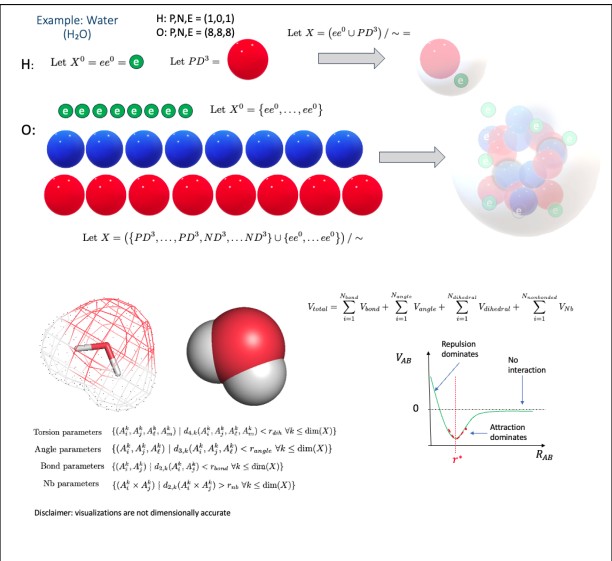

Figure 1: We graphically describe, with the example of $H_2O$, how to construct a Polyatomic complex. The first step is to encode each individual atom in detail (protons, neutrons, electrons). We view protons, neutrons and electrons as $n$-spheres. Afterward we combine our representations of each individual atom together to form molecules/atomistic systems. One can choose to further augment the representation at this step. Finally, one can feed the representation into a machine learning model to predict a property of interest.

**Graphical Abstract** In Figure 1 we graphically outline how to construct a polyatomic complex. In the first step we encode each individual atom by describing it as a particular kind of CW-complex, which we term an atomic complex. We discuss how to mathematically construct the atomic complex in Section 2. For every atom, we leverage its corresponding atomic complex and combine them to construct a CW-complex which we term a polyatomic complex. Upon constructing the geometric representation one may augment with additional features derived from force fields or molecule specific properties of interest. In our algorithm, by default, we compute a random matrix encoding pairwise forces and energetics, radial contribution. The resulting representation can be fed into a machine learning model. In practice, the final form of our representation is a PyTorch tensor. We elaborate on this in Section 2.3.

**SMILES** Simplified molecular-input line entry systems are obtained by printing symbol nodes encountered in a depth first traversal of a chemical graph (Weininger, 1988). SMILES are suitable for small molecules, but not large molecules. Additionally, they generate a substantial number of invalid molecules and fail to efficiently handle rings, branches, and bonds between atoms (Bhadwal et al., 2023).

**DeepSMILES** DeepSMILES was created to resolve some of the issues with SMILES. In particular, DeepSmiles addresses concerns with unmatched parentheses and ring closure symbols (O'Boyle & Dalke, 2018). However, DeepSMILES still allows for semantically incorrect strings (Krenn et al., 2022).

**ECFP Fingerprints** ECFP fingerprints are derived using a variant of the Morgan algorithm (Rogers & Hahn, 2010). Specifically, in the iterative updating stage, ECFP fingerprints encode the numbering invariant atom information to an atom identifier. During the duplicate removal stage, ECFPs reduce occurrences of the same feature (Rogers & Hahn, 2010). It is important to note that ECFPs are not ideal for substructure searching, as they are slow when applied to large databases. In addition, ECFP fingerprints are non-invertible because the hash function maps randomly to integers. Further, ECFPs do not contain the chemical properties of each atom (Le et al., 2020; Probst & Reymond, 2018; Xie et al., 2020).

**SELFIES** SELFIES follow a specific set of derivation rules, which enable the representation to satisfy semantic and syntactic constraints, while also avoiding syntactic mistakes (Krenn et al., 2020). SELFIES cannot fully represent macromolecules, crystals, and molecules with complicated bonds (Krenn et al., 2022). Similarly, SELFIES suffer from issues with substructure control, as described in Cheng et al. (2023).

**GroupSELFIES** GroupSELFIES are a more recent modification of SELFIES (Cheng et al., 2023). They remedy some of the issues with SELFIES by cleverly adding group tokens. Such an approach ensures that structures, like benzene rings, are more often preserved if shuffling occurs (Cheng et al., 2023). The primary limitation with GroupSELFIES is that groups cannot overlap. Consequently, one cannot represent polycyclic compounds (Cheng et al., 2023).

**Graphs** A graph $\mathcal{G}$ is an ordered pair $(\mathcal{V}, \mathcal{E})$, where $\mathcal{V}$ comprises a set of vertices and $\mathcal{E} \subset \mathcal{V} \times \mathcal{V}$ is composed of a set of edges (Griffiths et al., 2023). For molecules, the vertices $\mathcal{V} = \{v_1, \ldots, v_n\}$ represent the atoms of a molecule and the edges $\mathcal{E}$ represent covalent bonds between the atoms (Griffiths et al., 2023). Additional vertex and edge labels may be incorporated. The simplicity of graph representations poses some weaknesses, however. For example, there is no natural way to represent 3-dimensional structures of molecules and many other chemical properties that are essential to the molecule's functionality (Liu et al., 2022).

**Atomic Cluster Expansion (ACE)** Atomic Cluster Expansion is a technique which enables one to efficiently parameterize many atom interactions (Bochkarev et al., 2024). The basis functions of ACE are complete and can represent other local descriptors such as SOAP. In fields such as high energy physics, ACE is utilized to construct many-body interaction models which respect physical symmetries (Ho et al., 2024). However, standard ACE is known to be inefficient, thereby leading to numerous proposed variants with different trade-offs (Ho et al., 2024; Lysogorskiy et al., 2021a).

**Behler-Parrinello Descriptor** Behler-Parrinello is an ANN architecture that describes atomic environments with symmetry functions and relies on an element-specific neural network for atomic energies (Behler, 2015a). Behler-Parrinello belongs to a class of machine learning potentials wherein

the potential-energy surface (PES) is learned by adjusting parameters. The goal of fitting the PES is usually to accurately reproduce reference electronic structure data (Behler, 2015b).

**Bartók/SOAP Descriptor**   Smooth Overlap of Atomic Positions (SOAP), sometimes called the Bartók descriptor, is a technique to encode regions of atomic geometries (Barnard et al., 2023). SOAP relies on locally expanding a gaussian smeared atomic density with orthonormal functions (Barnard et al., 2023). The descriptor has been proven to be invariant to the basic symmetries of physics namely rotation, reflection, translation, and permutation of atoms of the same species Barnard et al. (2023).

We summarize our comparison of these methods by using tables found in the Appendix **??**.

## 1.1   Motivation and Representation Criteria

All representations listed above violate a non-empty subset of these criteria: invariances, uniqueness, continuity, differentiability, generality, computational efficiency, topological accuracy, ability to consider long-range interactions, and chemical informedness (Behler, 2011; Langer et al., 2022; Pozdnyakov et al., 2020; Todeschini & Consonni, 2008). In this work, we propose a new representation, polyatomic complexes, that mathematically satisfies the above constraints, consequently addressing the limitations of all discussed representations.

**Invariances**   We consider invariance under changes in atom indexing and those fundamental to physics. These invariances are rotation, reflection, and translations. We prove that polyatomic complexes satisfy all these invariances ( 2.12).

**Uniqueness**   As described by Langer et al. (2022), uniqueness refers to the idea that two systems differing in properties should be mapped to different representations. Systems with equal representations that differ in property induce errors. Uniqueness is necessary and sufficient for reconstruction, up to invariant transformations, of an atomistic system from its representation (Langer et al., 2022). We prove that polyatomic complexes satisfy uniqueness( 2.12).

**Continuity and Differentiability**   Representations of atomistic systems should be continuous and differentiable with respect to atomic coordinates (Langer et al., 2022). Moreover, discontinuities work against regularity assumptions of many machine learning models. We prove that polyatomic complexes are continuous and differentiable with respect to atomic coordinates ( 2.12).

**Generality**   We say a representation of atomistic systems or molecules is generalizable only if it can encode any atomistic system. SMILES, SELFIES, and GroupSELFIES are not generalizable, as they cannot represent certain kinds of molecule (crystals, polycyclic compounds, etc.) or they generate invalid molecules. However, we prove that polyatomic complexes are generalizable ( 2.12).

**Efficiency**   Traditionally, computational efficiency is a measure of how well an algorithm utilizes memory or time when completing a task. In the context of molecular representations, efficient representations run in polynomial time and with polynomial space. Ideally, representations are linear in the number of elements in a molecule, $O(S)$, as is the case with molecular graphs. In practice, polyatomic complexes are linear in the number of elements in a molecule, essentially $O(S)$. We provide a complete proof of time complexity in Theorem 2.12. However, in the case of Atomic Cluster Expansions (ACE), the algorithm described is not computationally efficient (Dusson et al., 2021). Traditionally ACE models are built using atomic properties $\Phi_i$ which are expanded in terms of body-ordered functions from the set of neighbors of each atom $i$. Classically, this leads to, for $\nu$ ordered basis-functions, $O(N^\nu)$ computational cost, where $N$ denotes the number of interacting neighbors (Lysogorskiy et al., 2021b). However, using the density trick leads to faster evaluation leading to the computational cost of an atomic property $\Phi_i$ scaling linearly in $N$ and also linearly in $\nu$ (Lysogorskiy et al., 2021b; Dusson et al., 2021). More efficient schemes such as PACE developed by Lysogorskiy et al. (2021b) avoid the $\nu$ scaling altogether, yet are still two orders of magnitude slower than empirical potentials (Lysogorskiy et al., 2021b). In the case of the Bartók/SOAP descriptor, we observe a rapid increase in descriptor size for environments composed of multiple elements (Darby et al., 2022). The SOAP power spectrum scales quadratically with the number of elements $S$, while the length of the bispectrum scales as $S^3$. As a result, SOAP descriptors are significantly less efficient than graph based methods which run in $O(S)$. As a result of the design of Behler-Parrinello, they are slower than $O(S)$ (Behler, 2015a). It should be noted that ACE, SOAP, and Behler-Parrinello are traditionally used for quantum-chemistry applications and are not designed for more typical computational chemistry tasks.

**Topological Accuracy**  A representation is deemed topologically accurate if it can correctly represent the geometry of any molecule or atomistic system. Correctness requires representing the shape, bond-angles, dihedrals/torsion, and electronic structure aspects accurately. Only polyatomic complexes, SOAP, and ACE are "reasonably" topologically accurate. Polyatomic complexes are not necessarily topologically accurate when it comes to electronic structure. Our representation, in practice, uses s-orbitals to represent electronic wave functions, which is an oversimplification. However, this can be altered at the cost of additional time complexity. Inclusion of a complete set of commuting observables (CSCO) would potentially enable perfect topological accuracy. SOAP enforces differentiability with respect to the atoms and invariance with respect to the basic symmetries of physics. Additionally, SOAP considers the potential energy surfaces (PESs) and electrostatic multipole moment surfaces. Similarly, polyatomic complexes enforce differentiability, are invariant to the basic symmetries of physics, and can be augmented with electronic structure aspects and forces. ACEs are invariant under the basic symmetries of physics and systematically describe the local environments of particles at any body order.

**Long-range interactions**  The term long-range interactions refers to electrostatic potential energies between atoms and molecules, with mutual distances ranging from a few tens to a few hundreds Bohr radii (Lepers & Dulieu, 2017). Interactions can only be evaluated up to a certain distance. The maximum distance applied in a simulation is usually referred to as the cut-off radius, $r_c$, because the Lennard-Jones potential is radially symmetric. Behler-Parinello neglects long-range interactions, which are electrostatics beyond the cutoff radius (Ko et al., 2021). Contrastingly, polyatomic complexes can adjust for varying definitions of cutoff radius (A.19). Similarly all string based representations and graph representations neglect long-range interactions. It should be noted that the precise definition of cutoff radius is dependent on the force field. The value of $r_c$ is generally obtained empirically.

**Chemical and Physical Informedness**  We say a representation is well-informed by chemistry or physics if it contains information about the chemical properties of each individual atom. ECFP fingerprints are not well-informed under this definition. If one compares ECFP fingerprints, Graphs, SMILES, or SELFIES to representations like ACE, SOAP, or polyatomic complexes, it is apparent that the former group does not contain the same level of chemical information as the latter. In essence, the representations should encode electronic structure, radial functions, spherical harmonics, wave-functions, and long-range interactions effectively. ACE utilizes basis functions to efficiently parameterize many-atom interactions (Qamar et al., 2023). SOAP relies on the local expansion of a Gaussian smeared atomic density with orthonormal functions (Barnard et al., 2023). Polyatomic complexes describe the geometry of individual atoms and encode radial functions in an efficient manner. They are flexible enough to potentially encode information about spherical harmonics as well. Additionally, they are compatible with force fields and can calculate the RDF as outlined in Section 2.3. It is theoretically shown that such an integration can be done. In practice, encoding spherical harmonics is left for future work.

In order to address the limitations of other representations, we develop Polyatomic Complexes. We propose a computationally efficient sampling-based approach for representing atomistic systems as CW-complexes. Our proposed sampling-based approach is computationally efficient and compatible with traditional physics-based methods. We delve into the mathematical construction in subsections 2.1 and 2.2. In subsection A.19, we discuss how our representation can integrate with techniques in chemistry and physics. The experiments can be found in section 3. Our conclusion is in section 4. All proofs of theorems, lemmas, and additional definitions are found in the appendix A.

## 1.2 CONTRIBUTIONS

Our theoretical contributions are as follows. We develop a representation that:

- satisfies all constraints outlined in Section 1.1.

- is generalizable (unlike SELFIES, SMILES, etc.).

- is simultaneously computationally efficient and can consider long-range interactions (unlike traditional representations in quantum chemistry).

- is geometrically and topologically accurate, regardless of the molecule/atomistic system one tries to model.

- is well-informed by physics and chemistry.

Contributions from our experiments are as follows. We provide/demonstrate:

- strong statistical evidence that our representation is competitive (in terms of accuracy) with the most commonly used representations in computational chemistry.
- our representation works well on crystal and materials datasets (Matbench, Materials Project) and smaller molecule datasets.
- a Gaussian process in tandem with our representation is competitive with GNN's (Matbench).

In summary, we develop a representation that provides theoretical guarantees (invariance with respect to the fundamental symmetries of physics, etc.) while being computationally efficient, competitive in terms of accuracy, and agnostic concerning the atomistic system one chooses to model.

## 2 CORE METHODS AND REPRESENTATION

In this section, we outline how we model protons, neutrons, and electrons. These assumptions are, to an extent, simplifications. We assume that an atom consists of a positively-charged nucleus, containing protons and neutrons, and is surrounded by negatively-charged electrons. Additionally, a proton $p$ is isomorphic to a sphere with radius $r \neq 1$. Similarly, a neutron $n$ is isomorphic to a sphere with radius $r \neq 1$. Further, an electron $e$ is isomorphic to a sphere and is paired up with a collapsed wave-function $w_e$, wherein the position is not definite; instead, the probability of finding an electron in a certain region can be calculated (Section 2.1). An atom, then, is a particular CW-complex formed by gluing together protons, neutrons, and electrons. We view protons, neutrons and electrons as objects in the category of topological spaces, and the gluing maps between spaces as an implicit force model.

### 2.1 MATHEMATICAL REPRESENTATION OF AN ATOM

We now develop formal mathematical definitions that can be used to reason about the objects we construct in our computational representation.

**Definition 2.1.** We define the set $ee^i := \{x \in \mathbb{R}^{i+1} \mid \|x\| < 2.8fm\}$ where $1fm = 1 \times 10^{-15} \ m$.

**Definition 2.2.** We define the set $PD^i := \{x \in \mathbb{R}^i \mid \|x\| \leq 1fm\}$ where $1fm = 1 \times 10^{-15} \ m$.

**Definition 2.3.** We define the set $ND^i := \{x \in \mathbb{R}^i \mid \|x\| \leq 0.8fm\}$ where $1fm = 1 \times 10^{-15} \ m$.

An atom is composed of protons, neutrons, and electrons. We construct protons, neutrons, and electrons using the sets $ee^i, PD^i$ and $ND^i$.

(1) Electrons: We know from above that an electron is isomorphic to a sphere and is paired up with a collapsed wave-function $w_e$. We know that $ee^i$ is a set and $w_e$ is a function. Functions can be represented as sets of ordered pairs $(x, y)$ such that $x \in X$ and $y \in Y$ when $w_e : X \to Y$. We can then let $Z = \{ee^i, \{(x, y) \mid x \in X, y \in Y\}\}$. Give $Z$ the indiscrete topology $\tau$. A single electron $e$ is then $(Z, \tau)$. Therefore electron $e \in \text{Top}$. Consequently, we can view $e$ as an object in the category of topological spaces.

(2) Protons: A proton is isomorphic to a sphere therefore we represent a single proton $p$ as a set $PD^i$ for $i \in \mathbb{N}$. Geometrically this set is a filled $i - 1$-sphere with a radius of $1fm$ for instance realized up to homeomorphism as a closed ball in Euclidean space.

(3) Neutrons: A neutron is isomorphic to a sphere therefore we represent a single neutron $n$ as a set $ND^i$ for $i \in \mathbb{N}$. Geometrically this set is a filled $i - 1$-sphere with a radius of $0.8fm$ for instance realized up to homeomorphism as a closed ball in Euclidean space.

It should be noted that one can change the values we have set for all the radii at will. We mention this because the exact radius of a proton is contested (Lin et al., 2022). We now proceed to develop our representation in a manner following the usual conventions in Algebraic Topology (Hatcher, 2002).

**Definition 2.4.** We define the **Atomic Complex** with A.17 in mind.
Suppose that atom $A$ has $\mathcal{N} \in \mathbb{N}$ many neutrons $\mathcal{P} \in \mathbb{N}$ many protons and $\mathcal{E} \in \mathbb{N}$ many electrons.

Let $I_n = \{1, \ldots, \mathcal{N}\}$, $I_p = \{1, \ldots, \mathcal{P}\}$, $I_e = \{1, \ldots, \mathcal{E}\}$ be index sets enumerating protons, neutrons and electrons respectively. Additionally we assert that $K = \mathcal{P} + \mathcal{N} + \mathcal{E}$. [1]

Let $\mathcal{T} = \{0, 1, \ldots, K\}$ such that $\mathcal{T}$ is ordered and $\tau_i \in \mathcal{T}$.

Then for any $\tau_i$ we can generalize A.17 by attaching many protons, neutrons and electrons.

We let $P$ be our complex of protons, $N$ be our complex of neutrons, $E$ our complex of electrons.

We then construct sets of attaching maps for each complex $\chi_p = \{\phi_{p,i} : \partial PD^{\tau_i} \to P_i \mid \forall i \in I_p\}$, $\chi_n = \{\phi_{n,i} : \partial ND^{\tau_i} \to N_i \mid \forall i \in I_n\}$, and $\chi_e = \{\phi_{e,i} : \partial ee^{\tau_i} \to E_i \mid \forall i \in I_e\}$.

We can think of these sets as continuous functions between the disjoint union spaces.

More formally we can write $(\phi_{p,i})_{i \in I_p} : \bigsqcup_{i \in I_p} \partial PD^{\tau_i} \to P_i$, $(\phi_{n,i})_{i \in I_n} : \bigsqcup_{i \in I_n} \partial ND^{\tau_i} \to N_i$, and $(\phi_{e,i})_{i \in I_e} : \bigsqcup_{i \in I_e} \partial ee^{\tau_i} \to E_i$.

Recalling from above we know that $PD^{\tau_i}, ND^{\tau_i}, ee^{\tau_i} \in \text{Top}$. Therefore we have three sequences of topological spaces $P = P_0 \hookrightarrow P_1 \hookrightarrow \cdots \hookrightarrow P_{\mathcal{P}}$, $N = N_0 \hookrightarrow N_1 \hookrightarrow \cdots \hookrightarrow N_{\mathcal{N}}$, and $E = E_0 \hookrightarrow E_1 \hookrightarrow \cdots \hookrightarrow E_{\mathcal{E}}$. [2]

For each $P_i \hookrightarrow P_{i+1}$, $N_i \hookrightarrow N_{i+1}$, and $E_i \hookrightarrow E_{i+1}$ we have that the following commute:

$$
\begin{array}{ccccc}
\bigsqcup_{i \in I_p} \partial PD^{\tau_i} \xrightarrow{(\phi_{p,i})_{i \in I_p}} P_i & \quad & \bigsqcup_{i \in I_n} \partial ND^{\tau_i} \xrightarrow{(\phi_{n,i})_{i \in I_n}} N_i & \quad & \bigsqcup_{i \in I_e} \partial ee^{\tau_i} \xrightarrow{(\phi_{e,i})_{i \in I_e}} E_i \\
\downarrow \qquad\qquad \downarrow & & \downarrow \qquad\qquad \downarrow & & \downarrow \qquad\qquad \downarrow \\
\bigsqcup_{i \in I_p} PD^{\tau_i} \longrightarrow P_{i+1} & & \bigsqcup_{i \in I_n} ND^{\tau_i} \longrightarrow N_{i+1} & & \bigsqcup_{i \in I_e} ee^{\tau_i} \longrightarrow E_{i+1}
\end{array}
$$

We combine the spaces $P$,$N$,$E$ together to get our atomic complex $A$. We let $\phi_p : \partial P \to \partial N$, $\phi_n : \partial N \to \partial E$, and $\phi_e : \partial E \to A$. Thus $A = A \cup_{\phi_p} P \cup_{\phi_n} N \cup_{\phi_e} E \in \text{Top}$. [3]

## 2.2 MATHEMATICAL REPRESENTATION OF POLYATOMIC SYSTEMS

In this section, we provide a generalized mathematical algorithm for modeling any polyatomic system. An atom, from this point onward, is represented by an atomic complex $A \in \text{Top}$ with the monad structure from $\text{Ato}$. We describe this structure in the Appendix A.20.

**Definition 2.5.** We define the **Polyatomic Complex** with A.5 and 2.4 in mind.

Suppose that atomistic system $M$ has $\mathcal{K}$ many atoms. Let $I_a = \{1, \ldots, \mathcal{K} - 1\}$ be an index set enumerating atoms. We can generalize A.5 by attaching $\mathcal{K}$ many atoms, the corresponding force field, and electronic structure spaces. We let $\phi_{A_{\mathcal{K}}} : \partial A_{\mathcal{K}} \to M$ and construct our set of attaching maps: $\chi_a = \{\phi_{a,i} : \partial A_i \to A_{i+1} \mid \forall i \in I_a\}$. We can think of these sets as continuous functions between the disjoint union spaces. In essence, $(\phi_{a,i})_{i \in I_a} : \bigsqcup_{i \in I_a} \partial A_i \to A_i$. Recalling from above, we know that $\forall i \in I_a$ that $A_i \in \text{Top}$. Therefore, we have a sequence of topological spaces: $M = A_0 \hookrightarrow A_1 \hookrightarrow \cdots \hookrightarrow A_{\mathcal{K}}$. [4]

$$
\begin{array}{ccc}
\bigsqcup_{i \in I_a} \partial A_i \xrightarrow{(\phi_{a,i})_{i \in I_a}} \partial A_{\mathcal{K}} \xrightarrow{\quad \phi_{A_{\mathcal{K}}} \quad} M \\
\downarrow \qquad\qquad\quad \downarrow \qquad\qquad\qquad \downarrow \\
\bigsqcup_{i \in I_a} A_i \longrightarrow A_{\mathcal{K}} \longrightarrow M = M \cup_{(\phi_{a,i})_{i \in I_a}} \left( \bigsqcup_{i \in I_a} A_i \right) \cup_{\phi_{A_{\mathcal{K}}}} A_{\mathcal{K}}
\end{array}
$$

Thus, $M = M \cup_{(\phi_{a,i})_{i \in I_a}} \left( \bigsqcup_{i \in I_a} A_i \right) \cup_{\phi_{A_{\mathcal{K}}}} A_{\mathcal{K}}$. [5]

*Remark* 2.6. It is apparent, at this point, that Polyatomic Complexes and Atomic Complexes are, by construction, finite $n$-connected CW-complexes. In any case, we prove each statement.

**Lemma 2.7.** *Atomic complexes are finite $n$-connected CW-complexes.*
*The full proof can be found in the Appendix A.6.*

**Lemma 2.8.** *Polyatomic complexes are finite $n$-connected CW-complexes.*
*The full proof can be found in the Appendix A.7.*

---

[1] We explicitly make a design choice to let $K = \mathcal{P} + \mathcal{N} + \mathcal{E}$ primarily for dimensionality reasons.

[2] The reasoning for why we need three separate complexes that we combine, is that $\mathcal{P}$ may not equal $\mathcal{N}$ which may not equal $\mathcal{E}$. Note that each of the inclusions $P_i \hookrightarrow P_{i+1}$, $N_i \hookrightarrow N_{i+1}$, and $E_i \hookrightarrow E_{i+1}$ induces an isomorphism on $\pi_i$ for all $i \le n$. Additionally note that $\pi_i$ denotes the $i$-th fundamental group.

[3] $A$ is initially the empty space $A = \varnothing$

[4] Note that each inclusion $A_i \hookrightarrow A_{i+1}$ induces an isomorphism on $\pi_i$ for $i \le n$.

[5] $M$ is initially the empty space $M = \varnothing$

**Theorem 2.9.** *Every Polyatomic complex has a smooth approximation.*
*The full proof can be found in the Appendix A.8.*

*Remark* 2.10. Note that the atomic coordinates are contained in a linear subspace of $\mathbb{R}^3$. Therefore, the atomic coordinates are a smooth embedded submanifold of $\mathbb{R}^3$ by A.15.

**Theorem 2.11.** *Atomic complexes are unique, continuous and differentiable with respect to atomic coordinates.*
*The proof for uniqueness can be found in the Appendix A.9. The proof of continuity and differentiability is found in the Appendix A.10.*

**Theorem 2.12.** *Polyatomic complexes satisfy all requirements for representations of atomistic systems, as outlined by Langer et al. (2022).*
*In essence, we want to prove invariances, uniqueness, continuity, differentiability, generality, and efficiency are all satisfied. All the aforementioned conditions are defined in Section 1.1. The full proofs for every condition are found in the Appendix.*

*Invariances: The full proof of satisfying all invariances can be found in the Appendix A.11.*

*Uniqueness: Uniqueness is necessary and sufficient for reconstruction, up to invariant transformations, of an atomistic system from its representation. The full proof of uniqueness can be found in the Appendix A.12.*

*Continuity and Differentiability: The full proof of continuity and differentiability with respect to atomic coordinates can be found in the Appendix A.13.*

*Generality: The full proof of generality can be found in the Appendix A.14.*

*Regressions: An important condition to demonstrate is that the structure of of our representation is suitable for regression. We provide statistical evidence of this in our experiments.*

*Computational efficiency: A proof of computational efficiency relative to the reference simulations is found in the Appendix A.15.*

### 2.3 ALGORITHMS

In this section, we discuss our implementation of the algorithm for Atomic and Polyatomic complexes. We provide the full pseudocode in the Appendix A.16. We refer to the pseudocode as Algorithm 1 and 2 respectively.

#### 2.3.1 ALGORITHMIC METHOD FOR ATOMIC COMPLEXES

In Algorithm 1, we provide our approach. In practice, we sample from the surface of each $n$-sphere uniformly when $i > 0$ for $PD^i, ND^i, ee^i$. In Algorithm 1, we let $D_F$ denote a random matrix encoding pairwise forces and energetics. $D_F$ may be derived from a known force field model or be randomly initialized (Brooks et al., 2009; Grimme, 2014; Senftle et al., 2016). The method $update\_distances$ uses information from the electron wave function to update distance information (radial contribution). The matrix $D_E$ keeps track of this distance information. The function $glue(T, o, \phi)$ glues object $o$ to space $T$ by using gluing map $\phi$. Gluing can be as simple as an append or as complex as one likes. Using a dictionary allows one to specify a gluing scheme.

#### 2.3.2 ALGORITHMIC METHOD FOR POLYATOMIC SYSTEMS

In Algorithm 2, we assume that all atoms are connected. This makes intuitive sense, as there are electromagnetic forces, strong nuclear forces, and weak nuclear forces present. Additionally, there are strong attractive forces, or intermolecular forces acting on all constituents. The functions $update\_radial$ and $update\_forces$ can optionally be provided by the user or default to standard behavior. If utilized, $update\_radial$ calculates the $RDF$ by picking a particular atom and calculating the density within the sphere. This can be implemented via Monte-Carlo methods as described by Lyubartsev & Laaksonen (1995). We use approximations of Algorithm 2 below.

**Fast Complex** In this approximation algorithm, we disconnect all cells in complex $C$ and project the entire representation onto the real plane to get a corresponding matrix $M \in \mathrm{GL}_n(\mathbb{R})$. We zero-pad

to ensure all matrices in our dataset have the same shape. We do not use any force field model, *update_forces* or *update_distances*.

**Deep Complex** In this approximation algorithm, we project all cells in $C$ onto the real plane to get a corresponding matrix $M \in \mathrm{GL}_n(\mathbb{R})$. We do, however, preserve connectedness information. We zero-pad to ensure all matrices in our dataset have the same shape. We do not use any force field model, *update_forces*, or *update_distances*.

## 3 BENCHMARKS AND EXPERIMENTATION

In each experiment, we do the following. First, for all representations, $n_{trials} = 20$, $n_{epochs} = 5$, and the train/test split is 67/33, except for Materials project where we use 10/90. Then, we process our dataset by converting SMILES to the representation being tested. We choose a kernel and fit an exact Gaussian process to our data. Upon completion, we report the average MAE, RMSE, and CRPS across all trials. We calculate 1-sigma error bars using bootstrapping, which represent the standard error across trials. For reference, choosing SELFIES as a representation with the Tanimoto Kernel is state-of-the-art for molecular learning tasks (Griffiths et al., 2023). See A.21 for the compute cost.

### 3.1 DATASET OVERVIEW

- **Photoswitches**: The Photoswitches dataset comprises of approximately four hundred photo-switchable molecules and associated chemical properties Griffiths et al. (2022). A photo-switchable molecule displays two or more isomeric forms accessible using light. Separating the electronic absorption bands of these isomers enables addressing a specific isomer and achieving high photostationary state (PSS) Griffiths et al. (2022). The dataset contains transition wavelengths and photophysical properties predicted using DFT.

- **ESOL**: The ESOL dataset contains approximately eleven hundred organic small molecules and their corresponding logarithmic aqueous solubility values Delaney (2004). Aqueous solubility is the maximum amount of a compound that can dissolve in a given volume of water at a specific temperature, and pressure. This is a key property to predict in areas such as drug design, and biochemistry.

- **FreeSolv**: The FreeSolv dataset contains approximately six hundred molecules and their corresponding hydration free energies Mobley & Guthrie (2014). Hydration free energy (HFE) is a physicochemical property of molecules describing how small molecules transfer between gas and water, or their relative populations in gas and water at equilibrium. HFE is a surrogate for performance in estimating protein-ligand binding free energy and has been used to assess and optimize the accuracy of non-bonded parameters in empirical force fields Mobley & Guthrie (2014).

- **ChEMBL/lipophilicity**: The lipophilicity dataset contains approximately four thousand compounds curated from the larger ChEMBL dataset along with their octanol/water distribution coefficient ($\log$ D at pH 7.4) Gaulton et al. (2011). The octanol/water distribution coefficient is used to determine the hydrophobicity (lipophilicity) of a chemical compound, thereby enabling the prediction of its environmental fate, and potential for bioaccumulation in organisms.

- **Materials Project**: The Materials Project dataset is a large open source dataset consisting of approximately one hundred seventy thousand complex materials along with a myriad of properties Jain et al. (2013). In this manuscript, we predict the equilibrium reaction energy and magnetization normalized vol. However, there are numerous electrochemical and thermodynamic properties one can predict.

- **Matbench - JDFT2D**: The Matbench benchmark consists of thirteen tasks that vary in size and contain data from DFT derived and experimentally derived sources. Important tasks supported include predicting optical, electronic, thermodynamic and tensile properties of a given material or crystal. In particular we select the JDFT2D task in which one must predict exfoliation energies from crystal structure. The exfoliation energy is defined as the amount of energy required to extract a two-dimensional sheet from the surface of a bulk material. This is a key property when determining the synthesizability of certain compounds.

## 3.2 PHOTOSWITCHES DATASET

The photoswitches dataset consists of photoswitchable molecules reported as SMILES, experimental transition wavelengths reported in nanometers, and DFT-computed transition wavelengths reported in nanometers among others (Griffiths et al., 2022). It should be noted that, for particular columns of the Photoswitches dataset, there are some missing values. Since we believe the values to be missing at random, we utilize mean imputation, instead of discarding experimental data. Additional tables are found in Appendix A.18.

Table 1: Photoswitches benchmark

| Kernel | Representation | Z isomer $\pi$-$\pi$* wavelength (nm) | | | DFT Z isomer $\pi$-$\pi$* wavelength (nm) | | |
|---|---|---|---|---|---|---|---|
| | | RMSE | MAE | CRPS | RMSE | MAE | CRPS |
| Tanimoto | SELFIES | $7.7 \pm 0.3$ | $3.9 \pm 0.1$ | 2.9 | $4.3 \pm 0.2$ | $1.1 \pm 0.1$ | 0.8 |
| Tanimoto | SMILES | $6.9 \pm 0.3$ | $3.4 \pm 0.1$ | 2.8 | $4.2 \pm 0.2$ | $1.0 \pm 0.1$ | 0.8 |
| Tanimoto | ECFP | $7.0 \pm 0.2$ | $3.3 \pm 0.1$ | 2.8 | $4.3 \pm 0.3$ | $1.1 \pm 0.1$ | 0.8 |
| Tanimoto | **Fast Complex** | $8.0 \pm 0.3$ | $2.9 \pm 0.1$ | 2.9 | $4.3 \pm 0.3$ | $0.9 \pm 0.1$ | 0.9 |
| Tanimoto | **Deep Complex** | $8.0 \pm 0.3$ | $2.9 \pm 0.1$ | 2.9 | $4.3 \pm 0.3$ | $0.9 \pm 0.1$ | 0.9 |
| Weisfeiler-Lehman | Graphs | $7.9 \pm 0.3$ | $3.6 \pm 0.1$ | 2.7 | $4.7 \pm 0.2$ | $1.3 \pm 0.1$ | 0.8 |

## 3.3 EXPERIMENT: ESOL

ESOL describes a simple method for estimating the solubility of a compound directly from its structure (Delaney, 2004). We convert all SMILES strings to all other representations prior to fitting as part of the data processing step. The predicted log solubility column contains theoretical values. The measured log solubility column contains values obtained through experimentation. We report all values rounded up to one decimal place and the confidence limits to two places.

Table 2: ESOL benchmark

| Kernel | Representation | Predicted log solubility (mol/L) | | | Measured log solubility (mol/L) | | |
|---|---|---|---|---|---|---|---|
| | | RMSE | MAE | CRPS | RMSE | MAE | CRPS |
| Tanimoto | SELFIES | $0.5 \pm 0.01$ | $0.4 \pm 0.01$ | 0.9 | $0.8 \pm 0.01$ | $0.6 \pm 0.01$ | 1.2 |
| Tanimoto | SMILES | $0.6 \pm 0.01$ | $0.4 \pm 0.00$ | 1.0 | $0.8 \pm 0.01$ | $0.6 \pm 0.00$ | 1.2 |
| Tanimoto | ECFP | $0.7 \pm 0.01$ | $0.5 \pm 0.01$ | 1.0 | $1.0 \pm 0.01$ | $0.8 \pm 0.01$ | 1.2 |
| Tanimoto | **Fast Complex** | $1.7 \pm 0.01$ | $1.3 \pm 0.01$ | 1.3 | $2.1 \pm 0.01$ | $1.7 \pm 0.01$ | 1.7 |
| Tanimoto | **Deep Complex** | $1.7 \pm 0.01$ | $1.3 \pm 0.01$ | 1.3 | $2.1 \pm 0.01$ | $1.7 \pm 0.01$ | 1.7 |
| WL | Graphs | $0.4 \pm 0.00$ | $0.3 \pm 0.02$ | 0.9 | $0.8 \pm 0.01$ | $0.6 \pm 0.01$ | 1.2 |

## 3.4 EXPERIMENT: FREESOLV

The Free Solvation database (FreeSolv) consists of experimental and calculated hydration free energies for small neutral molecules in water, along with molecular structures, input files, references, and annotations (Mobley & Guthrie, 2014). We convert all SMILES strings to all other representations prior to fitting as part of the data processing step. The experimental and calculated values correspond to hydration free energies reported in kcal/mol. The calculated column contains theoretical values. The experimental column contains values obtained through experimentation.

Table 3: FreeSolv benchmark

| Kernel | Representation | experimental (kcal/mol) | | | calculated (kcal/mol) | | |
|---|---|---|---|---|---|---|---|
| | | RMSE | MAE | CRPS | RMSE | MAE | CRPS |
| Tanimoto | SELFIES | $1.7 \pm 0.10$ | $1.2 \pm 0.03$ | 2.0 | $1.6 \pm 0.06$ | $1.2 \pm 0.05$ | 2.3 |
| Tanimoto | SMILES | $1.9 \pm 0.05$ | $1.4 \pm 0.02$ | 2.1 | $1.8 \pm 0.02$ | $1.3 \pm 0.02$ | 2.3 |
| Tanimoto | ECFP | $2.0 \pm 0.06$ | $1.4 \pm 0.02$ | 2.1 | $2.2 \pm 0.04$ | $1.5 \pm 0.02$ | 2.3 |
| Tanimoto | **Fast Complex** | $3.8 \pm 0.05$ | $2.8 \pm 0.03$ | 2.8 | $4.2 \pm 0.05$ | $3.1 \pm 0.04$ | 3.1 |
| Tanimoto | **Deep Complex** | $3.9 \pm 0.06$ | $2.8 \pm 0.04$ | 2.8 | $4.2 \pm 0.05$ | $3.1 \pm 0.04$ | 3.1 |
| WL | Graphs | $1.4 \pm 0.02$ | $1.0 \pm 0.01$ | 2.0 | $1.2 \pm 0.02$ | $0.8 \pm 0.01$ | 2.2 |

## 3.5 EXPERIMENT: CHEMBL/LIPOPHILICITY

ChEMBL is an Open Data database containing binding, functional, and ADMET information for a large number of drug-like bioactive compounds (Gaulton et al., 2011). We convert all SMILES

strings to all other representations prior to fitting as part of the data processing step. The experimental values contain log octanol–water partition coefficients $\log(K_{ow})$ for bioactive compounds.

Table 4: ChEMBL benchmark

| Kernel | Representation | experimental | | |
|---|---|---|---|---|
| | | RMSE | MAE | CRPS |
| Tanimoto | SELFIES | $0.9 \pm 0.01$ | $0.7 \pm 0.01$ | 0.7 |
| Tanimoto | SMILES | $0.7 \pm 0.01$ | $0.6 \pm 0.01$ | 0.7 |
| Tanimoto | ECFP | $0.7 \pm 0.01$ | $0.6 \pm 0.00$ | 0.7 |
| Tanimoto | **Fast Complex** | $1.2 \pm 0.01$ | $1.0 \pm 0.00$ | 1.0 |
| Tanimoto | **Deep Complex** | $1.2 \pm 0.01$ | $1.0 \pm 0.00$ | 1.0 |
| WL | Graphs | $0.7 \pm 0.01$ | $0.5 \pm 0.00$ | 0.7 |

## 3.6 MATERIALS PROJECT

Materials Project is a large database containing $\approx 173,000$ complex materials (Jain et al., 2013). We run our GP using a fixed batch size. The experimental values are equilibrium reaction energy reported in eV/atom and magnetization normalized vol. reported in $\mu$B. We utilize the entire Materials Project database, not a subset. Some example materials are $U_3(HO_5)_2$, $LuTaO_4$, and $Dy(SiPd)_2$, referenced by their materials project codes mp-626062, mp-5489, and mp-3301 respectively.

Table 5: Materials Project Benchmarks

| Quantity | Fast Complexes | | |
|---|---|---|---|
| | RMSE | MAE | CRPS |
| Equilibrium Reaction Energy (eV/atom) | $0.3296 \pm 0.0005$ | $0.1017 \pm 0.0005$ | $0.1017 \pm 0.0005$ |
| Total Magnetization Normalized Vol ($\mu$B) | $0.0196 \pm 0.0000$ | $0.0101 \pm 0.0000$ | $0.0101 \pm 0.0000$ |

## 3.7 MATBENCH: JDFT2D

Matbench: JDFT2D is a task in which one must predict exfoliation energies from crystal structure.

Table 6: Matbench: JDFT2D Benchmarks

| ML Model | Representation | Exfoliation Energy (meV/atom) | |
|---|---|---|---|
| | | RMSE | MAE |
| MODNet (v0.1.12) | Graph | 96.7332 | 33.1918 |
| SchNet | Graph | 111.0187 | 42.6637 |
| ALIGNN | Graph | 117.4213 | 43.4244 |
| GP + Tanimoto | **Fast Complexes** | 117.5536 | 60.8629 |
| GP + Tanimoto | **Deep Complexes** | 117.7818 | 60.8381 |
| Finder_v1.2 (structure-based) | Graph | 120.0917 | 46.1339 |
| CrabNet | Graph | 120.0088 | 45.6104 |
| MegNet | Graph | 129.3267 | 54.1719 |

## 4 DISCUSSION

Polyatomic complexes are a generalizable, topologically-informed method for encoding atomistic systems. Our results raise several important questions for future work. How can we develop better learning methods and kernels that operate on CW-complexes? Our results suggest that the Tanimoto kernel is not well-suited to our representation, as it does not consider geometric information. The primary limitation of our method is that the observed accuracy is only comparable with existing methods and not significantly better. Future work may involve investigating how to improve the accuracy we observe. Discovering the ideal balance of computational cost and accuracy when comparing a precise wave-function to an approximated representation is an exciting direction.

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

## A  APPENDIX

### A.1  MOLECULAR KERNELS FOR GAUSSIAN PROCESSES

In this section we provide some common kernels used with Gaussian processes for molecular machine learning tasks.

**Definition A.1.** Tanimoto: Let $d \in \mathbb{N}_{\geq 1}$ and $x, x' \in \{0, 1\}^d$ be binary vectors where $\|\cdot\|$ is the Euclidean norm and $\sigma_f^2$ is the scalar signal variance hyperparameter. Then define

$$k_{tanimoto} = \sigma_f^2 \cdot \frac{\langle x, x' \rangle}{\|x\|^2 + \|x'\|^2 - \langle x, x' \rangle}$$

**Definition A.2.** String Kernel: Let $S$ be a SMILES or SELFIES string, $d \in \mathbb{N}_{\geq 1}$ and $\phi : S \to \mathbb{R}^d$ be a bag-of-characters representation then define for any strings $S, S'$

$$k_{string}(S, S') = \sigma^2 \langle \phi(S), \phi(S') \rangle$$

**Definition A.3.** Let $\mathcal{G}$ be a graph domain and $\mathcal{H}$ be a reproducing kernel Hilbert space (RKHS) in which an inner product between $g, g' \in \mathcal{G}$ is well defined. Let $\phi_\lambda : \mathcal{G} \to \mathcal{H}$ where $\lambda$ is a kernel specific hyperpameter and $\sigma^2$ is a scale factor. then define

$$k_{graph}(g, g') = \sigma^2 \cdot \langle \phi_{\lambda(g)}, \phi_\lambda(g') \rangle_{\mathcal{H}}$$

## A.2 DEFINITIONS: CW-COMPLEXES

We utilize the standard definitions of CW-complexes as defined by Whitehead (1949).

**Definition A.4.** A cell complex $K$, or alternatively a *complex*, is a Hausdorff space which is the union of disjoint open cells $e, e^n, e_i^n$ subject to the condition that the closure $\bar{e}^n$ of each $n$-cell, $e^n \in K$ is the image of a fixed $n$-simplex in a map $f : \sigma^n \to \bar{e}^n$ such that

(1) $f|\sigma^n - \partial\sigma^n$ is a homeomorphism onto $e^n$

(2) $\partial e^n \subset K^{n-1}$, where $\partial e^n = f\partial\sigma^n = \bar{e}^n - e^n$ and $K^{n-1}$ is the $(n-1)$-section of $K$ consisting of all the cells whose dimension do not exceed $n - 1$.

**Definition A.5.** A complex $K$, can be described as closure finite $\iff K(e)$ is a finite subcomplex, for every cell $e \in K$. Moreover since $K(p) = K(e)$ if $p \in e$ this is equivalent to the condition that $K(p)$ is finite for each point $p \in K$ (Whitehead, 1949).

**Lemma A.6.** *If $L \subset K$ is a subcomplex and $e \in L$ then $L(e) = K(e)$. As a result any subcomplex of a closure finite complex is closure finite (Whitehead, 1949).*

**Definition A.7.** A complex $K$ has the weak topology $\iff$ a subset $X \subset K$ is closed provided $X \cap \bar{e}$ is closed for each cell $e \in K$ (Whitehead, 1949).

**Definition A.8.** A CW-Complex is a complex which is closure finite and has the weak topology (Whitehead, 1949).

---

**Algorithm 0** Construction of a Cell Complex or CW-Complex | Hatcher (2002)

---

Let $e^i := \{x \in \mathbb{R}^i \mid \|x\| < 1\}$
Let $D^i := \{x \in \mathbb{R}^i \mid \|x\| \leq 1\}$
Let $S^{i-1} := \{x \in \mathbb{R}^i \mid \|x\| = 1\}$
Start with a set of points $K^0$
$(n = 1)$ Build $K^1$ by attaching the boundary of the 1-cell $e^1$ to $K^0$
$(n = 2)$ Build $K^2$ by attaching the boundary of the 2-cell $e^2$ to $K^1$
General case $(n = j)$: Build $K^j$ by attaching the boundary of the $n$-cell to $K^{j-1}$

Remark: Attaching a boundary means $\exists\varphi_\alpha : S^{j-1} \to K^{j-1}$ such that $K^j \leftarrow \frac{K^{j-1} \coprod_\alpha D_\alpha^j}{k \sim \varphi_\alpha(k)}$

---

In essence one is inductively forming the $n$-skeleton $K^n$ from $K^{n-1}$ by attaching $n$-cells via attaching maps $\varphi_\alpha : S^{j-1} \to K^{j-1}$. Therefore $K^n$ is the quotient space of the disjoint union $K^{n-1} \coprod_\alpha D_\alpha^n$ of $K^{n-1}$ with a collection of $n$-disks $D_\alpha^n$ under identifications $k \sim \varphi_\alpha(k)$ for $k \in \partial D_\alpha^n$. Thus $K^n = K^{n-1} \coprod_\alpha e_\alpha^n$. One can stop the induction at a finite state setting $K = K^n$ for $n < \infty$ or one can continue indefinitely setting $K = \bigcup_n K^n$. In the second case $K$ has the weak topology (Hatcher, 2002).

**Definition A.9.** Given a CW-Complex $X$, one denotes the $j$-th cell of dimension $k$ as $e_j^k$. Traditionally, one lets the relation $\prec$ denote incidence (Sardellitti & Barbarossa, 2024). If two cells $e_j^{k-1}$ and $e_i^k$ are incident, we write $e_j^{k-1} \prec e_i^k$. For not incident, we write $e_j^{k-1} \nprec e_i^k$. Let the relation $\sim$ denote orientation. We write $e_j^{k-1} \sim e_i^k$ if the cells have the same orientation. For the opposite orientation we write $e_j^{k-1} \nsim e_i^k$.

**Definition A.10.** The Hodge Laplacian $\Delta_k : C^k(X) \to C^k(X)$ on the space of $k$-cochains is then $\Delta_k := d_{k-1} \circ d_{k-1}^* + d_k^* \circ d_k$. The matrix representation is then $\Delta_k := B_k^\top W_{k-1}^{-1} B_k W_k + W_k^{-1} B_{k+1} W_{k+1} B_{k+1}^\top$. Here, $W_k = \mathrm{diag}(w_1^k, \ldots, w_{N_k}^k)$ is the diagonal matrix of cell weights and $B_k$ is the order $k$ incidence matrix, whose $j$-th column corresponds to a vector representation of the cell boundary $\partial e_j^k$ viewed as a $k-1$ chain (Khorana, 2024).

## A.3 THEOREMS: MANIFOLDS AND EMBEDDINGS

**Theorem A.11.** *(Whitney) The strong Whitney Embedding Theorem states that any smooth real $m$-dimensional manifold (Hausdorff and second-countable) can be smoothly embedded in $\mathbb{R}^{2m}$, if $m > 0$ (Whitney, 1944).*

**Theorem A.12.** *(Whitney) The weak Whitney Embedding Theorem states that any continuous function from an $n$-dimensional manifold to an $m$-dimensional manifold may be approximated by a smooth embedding provided $m > 2n$. Moreover, such a map can be approximated by an immersion provided $m > 2n - 1$ (Whitney, 1944).*

**Theorem A.13.** *(Lazarus) Any finite simplicial complex of dimension $n$ embeds linearly into $\mathbb{R}^{2n+1}$. This follows from the Whitney embedding theorem extended to simplicial complexes. We reproduce the full proof given by Lazarus (Lazarus, 2020).*

*Proof.* Define a linear mapping $f$ of the $n$-dimensional complex $K$ into $\mathbb{R}^{2n+1}$ by mapping the vertices of $K$ to points in general position in $\mathbb{R}^{2n+1}$, such that no hyperplane contains more than $2n + 1$ points. Then $f$ is an embedding. This is apparent when restricted to any simplex of $K$: the simplex has at most $n + 1$ vertices which are sent to affinely independent points by the general position assumption. To see that $f$ is injective we need to prove that distinct simplicies have their interior sent to disjoint sets. So let $\sigma = (v_1, \ldots, v_k)$ and $\tau = (w_1, \ldots, w_\ell)$ be two distinct simplicies of $K$. Since $k + \ell \leq 2n + 2$, the general position assumption implies that the image points $f(v_1), \ldots, f(v_k), f(w_1), \ldots, f(w_\ell)$ span a simplex of dimension $k + \ell - 1$ and that $f(\sigma)$ and $f(\tau)$ are two distinct faces of this simplex. It follows that $f(\tau)$ and $f(\sigma)$ are disjoint, Then by injectivity we have embedding for finite simplicial complexes. $\square$

**Definition A.14.** (Fushida) If $f : \mathbb{R}^n \to \mathbb{R}$ is a smooth function then its gradient is the vector field $\mathrm{grad}\, f$ defined by $\mathrm{grad}_x f = \left( \frac{\partial f}{\partial x_1}(x), \cdots, \frac{\partial f}{\partial x_n}(x) \right)$. Equivalently the vector field is defined by $g(\mathrm{grad} f, Y) = df(Y)$ for all vector fields $Y$ (Fushida-Hardy).

**Definition A.15.** A subset $M$ of linear space $E$ of dimension $d$ is a smooth embedded submanifold of dimension $n$ if $\forall x \in M$ there exists neighborhood $U$ of $x$ in $E$ and open set $V \subseteq \mathbb{R}^d$ and diffeomorphism $\varphi : U \to V$ such that $\varphi(U \cap M) = V \cap E$ where $E$ is a linear subspace of dimension $n$ (Lee & Lee, 2012).

*Remark* A.16. Rotations, reflections, and translations are coordinate transformations (Gower & Dijksterhuis, 2004).

## A.4 EXAMPLE: ATOMIC COMPLEX CONSTRUCTION FOR DEUTERIUM

**Lemma A.17.** *For the sake of illustration we describe a simple case, namely how we encode deuterium which contains 1 proton, 1 neutron, and 1 electron. Let* Top *be the category of topological spaces.*
*Let $p \cong PD^3$ be a proton, $n \cong ND^3$ be a neutron and $e = (Z, \tau)$ be an electron. Then we know that $p, n, e, \partial p, \partial n, \partial e \in$ Top. Let $\varphi_p : \partial p \hookrightarrow p$, $\varphi_n : \partial n \hookrightarrow n$, $\varphi_e : \partial e \hookrightarrow e$ be the generating cofibrations for protons, neutrons and electrons. Let the $n$-cell attachment to space $X$ be the result of gluing either $p, n$, or $e$ along a prescribed image of it's boundary. In essence $\phi_p : \partial p \to \partial n$, $\phi_n : \partial n \to \partial e$, $\phi_e : \partial e \to X$ and all are continuous functions. The attaching space is then $X \cup_{\phi_p} p \cup_{\phi_n} n \cup_{\phi_e} e \in$ Top which makes the following diagram commute:*

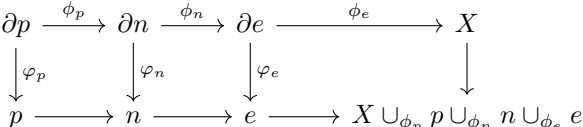

## A.5 Example: Polyatomic Complexes (two-atoms)

**Lemma A.18.** *For the sake of illustration we describe the simplest possible case of connecting 2 atoms. Let $A_1$ and $A_2$ be atoms. We know that $A_1, A_2 \in$ Top. Let $\varphi_{A_1} : \partial A_1 \hookrightarrow A_1$, $\varphi_{A_2} : \partial A_2 \hookrightarrow A_2$ be the generating cofibrations for each complex. Let the $n$-cell attachment to space $X$ be the result of gluing either $A_1$, or $A_2$, along a prescribed image of its boundary. In essence $\phi_{A_1} : \partial A_1 \to \partial A_2$, $\phi_{A_2} : \partial A_2 \to \partial X$ such that all are continuous functions.*
*The attaching space is then $X \cup_{\phi_{A_1}} A_1 \cup_{\phi_{A_2}} A_2$ which makes the following diagram commute:*

$$
\begin{array}{ccc}
\partial A_1 \xrightarrow{\phi_{A_1}} \partial A_2 \xrightarrow{\quad\phi_{A_2}\quad} & X \\
\downarrow{\varphi_{A_1}} \qquad \downarrow{\varphi_{A_2}} & \downarrow \\
A_1 \longrightarrow A_2 \longrightarrow X \cup_{\phi_{A_1}} A_1 \cup_{\phi_{A_2}} A_2
\end{array}
$$

$$
(f \circ g)(a_1)(Z) = \int_{A_2} g_{a_2}(Z) \, df_{a_1}(a_2)
$$

## A.6 Proof: Lemma 2.7

*Proof.* We want to show atomic complex $A$ is finite.
By 2.4 an atomic complex is a union of $P_i, N_j$, and $E_k$. The frequency of each is governed by $\mathcal{N}, \mathcal{P}, \mathcal{E}$ which correspond to the number of protons, neutrons and electrons found in an atom. Since individual atoms have a finite number of protons neutrons and electrons, we know each cell is finite and there are finitely many cells in each dimension. Therefore $A$ must be finite.
We want to show that an atomic complex $A$ is $n$-connected.
For every $i \in I_a$, let $g_i$ be a map from a one point space to $P_i$ or $N_j$ or $E_k$. Since both spaces have all homotopy groups trivial, $\forall i$ $g_i$ induces an isomorphism on all the homotopy groups. Therefore by Whitehead's Theorem for every $i$ we know $g_i$ is a homotopy equivalence and so $P$ is contractible $\implies P$ is $n$-connected.
Thus atomic complex $A$ is finite and $n$-connected. $\qquad\square$

## A.7 Proof: Lemma 2.8

*Proof.* We want to show polyatomic complex $P$ is finite.
By 2.5 an polyatomic complex is a union of atomic complexes $A_i$. The frequency of atomic complexes is governed by $\mathcal{K}$ which corresponds to the number of atoms in the system. Since the atomistic systems we model are finite, we know each cell is finite and there are finitely many cells in each dimension. Therefore $P$ must be finite.
We want to show that an polyatomic complex $P$ is $n$-connected.
For every $i \in I_a$, let $g_i$ be a map from a one point space to $A_i$. Since both spaces have all homotopy groups trivial, $\forall i$ $g_i$ induces an isomorphism on all the homotopy groups. Therefore by Whitehead's Theorem for every $i$ we know $g_i$ is a homotopy equivalence and so $P$ is contractible $\implies P$ is $n$-connected.
Thus polyatomic complex $P$ is finite and $n$-connected. $\qquad\square$

## A.8 Proof: Theorem 2.9

*Proof.* Let $P$ be a polyatomic complex. By 2.8 $P$ is a finite CW complex.
It is then implied that each $P$ is homotopy equivalent to a finite dimensional locally finite simplicial complex $S$ (Theorem 2C.5 Hatcher (Hatcher, 2002)). By A.13, we know $S$ has linear embedding $X \in \mathbb{R}^k$. Pick regular neighborhood $M_x$ of $X \in \mathbb{R}^k$. It will be homotopy equivalent to $X$ and a complex PL submanifold with boundary. Take the interior $\text{int}(M_x) = \mathcal{M}$. Then, $\mathcal{M}$ is a smooth manifold. $\qquad\square$

## A.9 Proof: Theorem 2.11 Uniqueness

*Proof.* We want to show that an atomic complexes are unique. In essence two systems differing in property should be mapped to different representations.

This immediately follows from Hatcher Corollary 4.19 (Hatcher, 2002), an $n$-connected CW model is unique up to homotopy equivalence rel$A$. By lemma 2.8 a polyatomic complex $P$ is $n$-connected.[6]

$\square$

## A.10 PROOF: THEOREM 2.11 CONTINUITY AND DIFFERENTIABILITY

*Proof.* We want to show that atomic complexes are continuous and differentiable with respect to atomic coordinates.

We want to show that atomic complexes are continuous with respect to atomic coordinates. Let $\mathcal{M}_A$ be the embedded smooth submanifold of atomic coordinates as in 2.10. Similarly let the smooth manifold $\mathcal{M}_C$ correspond to an atomic complex $C$ as in 2.9. Claim: The map $F : \mathcal{M}_C \to \mathcal{M}_A$ is smooth and therefore continuous. It is sufficient to show that $F$ is locally continuous i.e. $\forall x \in \mathcal{M}_C \exists$ neighborhood $U_x$ such that $F|_{U_x}$ is continuous. Then since both $\mathcal{M}_A$ and $\mathcal{M}_C$ are smooth manifolds there exist local coordinate systems $(U, \xi)$ at $x$ and $(V, \chi)$ at $F(x)$ such that the coordinate expression $\Phi = \chi \circ F \circ \xi^{-1} : \xi(U \cap F^{-1}(V)) \to \mathbb{R}^n$ is smooth. Pick $U_x = U \cap F^{-1}(V)$. Then it follows that $U_x \subset F^{-1}(V) \implies F(U_x) \subset F(F^{-1}(V)) \subset V$. Thus the maps $\xi^{-1}$ and $\chi$ are homeomorphisms and $\Phi$ is a smooth map between euclidean spaces and $F|_{U_x}$ is a composition of continuous maps $\implies F$ is continuous. Then let $g : C \to \mathcal{M}_C$ be continuous and $h : M_A \to A$ be continuous. Then composition of continuous functions are continuous $g \circ F \circ h$. Therefore we have continuity.

We want to show that atomic complexes are differentiable with respect to atomic coordinates. Let $\mathcal{M}_A$ be the embedded smooth submanifold of atomic coordinates as in 2.10. Similarly let the smooth manifold $\mathcal{M}_C$ correspond to an atomic complex $C$ as in 2.9. Then the differential of the map $F : \mathcal{M}_C \to \mathcal{M}_A$ is well defined. The differential of smooth map $F$ at $x$ $dF(x) : T_x\mathcal{M}_C \to T_{F(x)}\mathcal{M}_A$ is defined by: $dF(x)[v] = (F \circ c)'(0) = \lim_{t \to 0} \frac{F(c(t)) - F(x)}{t}$ where $c : \mathbb{R} \to \mathcal{M}_C$ satisfies $c(0) = x$ and $c'(0) = v$. This map is linear and we have a chain rule. If $\bar{F}$ is the smooth extension of $F$, then $dF(x) = d\bar{F}(x)|_{T_x\mathcal{M}_C}$. In general by A.14 we just write $g(\mathrm{grad}F, X) = dF(X)$ for vector field $X$. We can let $dF(x)$ be our (approximate) derivative for $C$ with respect to atomic coordinates. We can alternatively let $f : C \to \mathcal{M}_C$ be a continuous differentiable map, let $g : \mathcal{M}_A \to A$ be continuous and differentiable, and rely on chain rule. Therefore we can define the derivative with respect to atomic coordinates. $\square$

## A.11 PROOF: THEOREM 2.12 INVARIANCES

*Proof.* We want to show polyatomic complexes are invariant with respect to rotations, changes in atom indexing, translations and reflections.

It suffices to choose an atlas such that all the coordinate change maps are smooth. Let the smooth manifold $\mathcal{M}_C$ correspond to an polyatomic complex $C$ as in 2.9. Additionally, let $C^\infty([a, b])$ denote the space of smooth functions $f : [a, b] \to \mathcal{M}_C$. Then, for functionals $S : C^\infty([a, b]) \to \mathbb{R}$ of the form $S[f] = \int_a^b (L \circ \dot{f})(x)dx$ where $L : T\mathcal{M}_C \to \mathbb{R}$ is the Lagrangian, we know $dS_f = 0 \iff \forall t \in [a, b]$ each coordinate frame trivialization $(x^i, X^i)$ of a neighborhood of $\dot{f}(t)$ yields the following $\dim \mathcal{M}_C$ equations: $\forall i : \frac{d}{dx}\frac{\partial L}{\partial X^i}|_{\dot{f}(x)} = \frac{\partial L}{\partial x^i}|_{\dot{f}(x)}$. This recovers the Euler-Lagrange equations [7] $\mathcal{L}\theta_L = dL$ which are invariant under coordinate transformations and therefore rotationally, reflectionally, and translationally invariant. [8]

We want to show polyatomic complexes are invariant w.r.t changes in atom indexing.

Let $I = \{1, \ldots, i, i+1, \ldots, n\}$ and $n \in \mathbb{N}$ s.t. $1 \le i < i+1 \le n$. Suppose that polyatomic complex $P$ is composed of atomic complexes $[A_1, \ldots, A_i, A_{i+1}, \ldots, A_n]$. Suppose that we swap $A_i$ and $A_{i+1}$ and construct polyatomic complex $O = [A_1, \ldots, A_{i+1}, A_i, \ldots, A_n]$. Then, it suffices to show that $O = P$. We know that $O$ and $P$ are homeomorphic with isomorphic fundamental groups. In essence $\forall i$ we have $\pi_i(P) \xrightarrow{\sim} \pi_i(O)$. Since polyatomic complexes are contractible $\implies$ all homotopy groups are trivial $\implies \forall i \, \pi_i(P) = \pi_i(O) = 0$. Thus $O = P$. In essence, they encode the same information regardless of whether one re-indexes. $\square$

---

[6] All chemical elements are distinguished by their number of protons. In the case of isotopes the number of neutrons would differ. You would have a different number of cells.

[7] We can use the lie derivative $\mathcal{L}$ and local charts $(q^\alpha, \dot{q}^\alpha)$ such that $\theta_L = \frac{\partial L}{\partial \dot{q}^\alpha}dq^\alpha$ and $\Delta = \frac{d}{dx}$ (José & Saletan, 1998).

[8] A change of coordinates is a diffeomorphism between manifolds.

## A.12 Proof: Theorem 2.12 Uniqueness

*Proof.* We want to show polyatomic complexes are unique.

By Hatcher Corollary 4.19 (Hatcher, 2002), an $n$-connected CW model is unique up to homotopy equivalence rel$A$. By lemma 2.8, a polyatomic complex $P$ is $n$-connected.

□

## A.13 Proof: Theorem 2.12 Continuity and Differentiability

*Proof.* We want to show that polyatomic complexes are continuous and differentiable with respect to atomic coordinates.

We want to show that polyatomic complexes are continuous with respect to atomic coordinates. Let $\mathcal{M}_A$ be the embedded smooth submanifold of atomic coordinates as in 2.10. Similarly, let the smooth manifold $\mathcal{M}_C$ correspond to an polyatomic complex $C$ as in 2.9. Claim: The map $F : \mathcal{M}_C \to \mathcal{M}_A$ is smooth and therefore continuous. It is sufficient to show that $F$ is locally continuous i.e. $\forall x \in \mathcal{M}_C \exists$ neighborhood $U_x$ such that $F|_{U_x}$ is continuous. Then since both $\mathcal{M}_A$ and $\mathcal{M}_C$ are smooth manifolds there exist local coordinate systems $(U, \xi)$ at $x$ and $(V, \chi)$ at $F(x)$ such that the coordinate expression $\Phi = \chi \circ F \circ \xi^{-1} : \xi(U \cap F^{-1}(V)) \to \mathbb{R}^n$ is smooth. Pick $U_x = U \cap F^{-1}(V)$. Then, it follows that $U_x \subset F^{-1}(V) \implies F(U_x) \subset F(F^{-1}(V)) \subset V$. Thus, the maps $\xi^{-1}$ and $\chi$ are homeomorphisms and $\Phi$ is a smooth map between euclidean spaces and $F|_{U_x}$ is a composition of continuous maps $\implies F$ is continuous. Then we see $g : C \to \mathcal{M}_C$ and $h : \mathcal{M}_A \to A$ are continuous, since they are smooth embeddings. Essentially, we know $g$ is continuous on the restriction to every space in the diagram $\implies$ by the universal property of a colimit that $g$ is continuous. Then, composition of continuous functions are continuous. Therefore, we have continuity.

We want to show that polyatomic complexes are differentiable with respect to atomic coordinates. Let $\mathcal{M}_A$ be the embedded smooth submanifold of atomic coordinates as in 2.10. Similarly, let the smooth manifold $\mathcal{M}_C$ correspond to an polyatomic complex $C$ as in 2.9. Then the differential of the map $F : \mathcal{M}_C \to \mathcal{M}_A$ is well defined. The differential of smooth map $F$ at $x$ $dF(x) : T_x\mathcal{M}_C \to T_{F(x)}\mathcal{M}_A$ is defined by: $dF(x)[v] = (F \circ c)'(0) = \lim_{t \to 0} \frac{F(c(t)) - F(x)}{t}$ where $c : \mathbb{R} \to \mathcal{M}_C$ satisfies $c(0) = x$ and $c'(0) = v$. This map is linear and we have a chain rule. If $\bar{F}$ is the smooth extension of $F$, then $dF(x) = d\bar{F}(x)|_{T_x\mathcal{M}_C}$. In general, by A.14, we just write $g(\text{grad}F, X) = dF(X)$ for vector field $X$. We can let $dF(x)$ be our (approximate) derivative for $C$ with respect to atomic coordinates. Alternatively, we know $f : C \to \mathcal{M}_C$, and $g : \mathcal{M}_A \to A$ are smooth embeddings $\implies$ the derivatives are everywhere injective. Thus, we can compose these functions and apply chain rule. Therefore, we can define the derivative with respect to atomic coordinates. □

## A.14 Proof: Theorem 2.12 Generality

*Proof.* Suppose for the sake of contradiction that a polyatomic complex cannot encode any atomistic system. This implies, by definition an atomistic system, that there exists a finite collection of atoms $\{A_1, \ldots, A_k\}$ such that $\nexists$ a corresponding polyatomic complex $P$. However, we can construct such a $P$. By 2.4 $\forall A_i \in \{A_1, \ldots, A_k\}$ there exists corresponding atomic complex $\mathcal{A}_i$. Let our corresponding set of atomic complexes be $\{\mathcal{A}_1, \ldots, \mathcal{A}_k\}$. Then, can define attaching maps $\chi_a = \{\phi_{a,i} : \partial\mathcal{A}_i \to \mathcal{A}_i \mid a \text{ corresponds to } A_i\}$. Then, we have a sequence of topological spaces $\mathcal{A}_0 \hookrightarrow \ldots \hookrightarrow \mathcal{A}_K$ and can form $P$ such that a diagram $D$ isomorphic to the one in 2.5 commutes. Therefore, we have a valid polyatomic complex $P$. As a result, we have a contradiction and the opposite must be true. □

## A.15 Proof: Theorem 2.12 Time complexity

*Proof.* Suppose the reference simulation is a system with $k = |M_A|$ many atoms. By Algorithm 2, we can show that we can construct a polyatomic complex in $O(C)$ which is polynomial time.

For an atomic complex, Algorithm 1 runs in $O(M \cdot (d_n \cdot N + d_e \cdot E \cdot S + d_p \cdot P)) = O(A)$ where $P, N, E$ are the number of protons, neutrons, and electrons and $d_p, d_n, d_e$ correspond to the range of dimensions for protons, neutrons, and electrons; additionally, $S$ is the time complexity of the method in (Motta & Zhang, 2018), and $M$ is the time complexity of constructing an attaching map via helper function. Then, we know that Algorithm 2 runs in $O(k \cdot A \cdot F \cdot M \cdot R) = O(C)$ where $M$ is the

time it takes to construct 1 attaching map, $F$ is the time complexity of $update\_forces$, and $R$ is the time complexity of $update\_radial$. Thus, Algorithm 2 runs in $O(C)$. $\square$

## A.16 ALGORITHMS

In this section, we provide the pseudo-code for Atomic and Polyatomic complexes.

---

**Algorithm 1** Atomic Complexes

---

Let $A$ be an atom.
**Input:** Let $P$ be the number of protons, $E$ be the number of electrons, and $N$ be the number of neutrons. Pick a desired maximum dimension $K$ based on what one wants $\tau_i$ to be. Note that $K = |\mathcal{T}| = P + N + E$ as in definition 2.4. By default we fix $\tau_i = 3$ for protons and neutrons and $\tau_i = 0$ for electrons. If one wants to form complexes over a range of dimensions $[0, d]$, run each while loop $O(d)$ times and ensure $|A_E| = E \cdot d$, $|A_P| = P \cdot d$, $|A_N| = N \cdot d$.
Let $A_E := \{[ee_1^0, w_1], \ldots [ee_E^0, w_E]\}$ such that $|A_E| = E$.
Let $A_P := [PD^{\tau_i}, \ldots, PD^{\tau_i}]$ such that $|A_P| = P$.
Let $A_N := [ND^{\tau_i}, \ldots, ND^{\tau_i}]$ such that $|A_N| = N$.
Let $D_F$ denote a random matrix encoding pairwise forces and energetics between all protons and neutrons.
Let $D_E$ denote a random matrix encoding radial contribution for electrons.
Note that in general $\phi_{t,k} : \partial T_{n_k} \to T_k$ for all $k \in I_t$ are the attaching maps for protons, neutrons and electrons respectively.
Let $P_0 = \varnothing$
Let $i_p = 0$.
**while** $|A_P| > 0$ **do**
    Let $p = A_P.\text{pop}()$
    $P_{i_p} \leftarrow glue(P_{i_p-1}, p, \phi_{\tau_i, i_p})$
    $i_p = i_p + 1$
**end while**
Let $N_0 = \varnothing$
Let $i_n = 0$
**while** $|A_N| > 0$ **do**
    Let $n = A_N.\text{pop}()$
    $N_{i_n} \leftarrow glue(N_{i_n-1}, n, \phi_{\tau_i, i_n})$
    $i_n = i_n + 1$
**end while**
Let $E_0 = \varnothing$
Let $i_e = 0$
**while** $|A_E| > 0$ **do**
    Let $e, w_{i_e+1} = A_E.\text{pop}()$
    $E_{i_e} \leftarrow glue(E_{i_e-1}, e, \phi_{\tau_i, i_e})$
    $update\_distances(D_e, w_{i_e+1}, e)$
    $i_e = i_e + 1$
**end while**
Note that $\varphi_n : \partial P_{i_p} \to \partial N_{i_n}$ and $\varphi_p : \partial N_{i_n} \to \partial E_{i_e}$.
$K = P_{i_p} \cup_{\varphi_n} N_{i_n} \cup_{\varphi_e} E_{i_e}$
**return** $A = (K, A_E, D_F, D_E)$

---

**Algorithm 2** Polyatomic Complexes

---

Let $P$ be a polyatomic complex.
**Input:** Let $M_A$ be a list of atoms present in the system. Let $using\_radial$ and $using\_force\_model$ be boolean values.
// Note that for every $a \in M_A$ $a := (p, n, e, d)$ corresponding to number of protons, neutrons and electrons and desired dimensions or range of dimensions.
Let $\mathcal{A} \leftarrow List()$ be an empty dynamic array.
**for all** $a \in M_A$ **do**
    $A \leftarrow \text{Algorithm1}(a)$
    $append(\mathcal{A}, A)$
**end for**
Let $C = \varnothing$
Let $E = \text{matrix}(0, 0)$
Let $F = \text{matrix}(0, 0)$
Let $D_E = \text{matrix}(0, 0)$
Let $i = 0$
Note that $\phi_{a,i} : \partial K_i \to K_{i+1}$ for $i \in I_a$ are the attaching maps for the zeroth element of the atomic complex result in essence $A[0] = K$.

**while** $|\mathcal{A}| > 0$ **do**
    $k, a_e, d_f, d_e = \mathcal{A}.\text{pop}()$
    $C^i \leftarrow glue(C^{i-1}, k, \phi_{i,a})$
    $E = E \oplus a_e$
    **if** $using\_force\_model$ **then**
        $F = F \oplus d_f$
        $update\_forces(F)$
    **end if**
    **if** $using\_radial$ **then**
        $D_E = D_E \oplus d_e$
        $update\_radial(D_E)$
    **end if**
    $i = i + 1$
**end while**
**return** $P = (C^i, E, F, D_E)$

---

## A.17 METHOD COMPARISON:

We summarize the differences between methods in the following tables.

Table 1: Method Comparison I

| | Invariance | Uniqueness | Continuity & Differentiability | Generalizability | Efficiency |
|---|---|---|---|---|---|
| Polyatomic Complex | ✓ | ✓ | ✓ | ✓ | ✓: $O(S)$ |
| SMILES | ✗ | ✓ | ✓ [9] | ✗ | ✓ |
| SELFIES | ✗ [10] | ✓ [11] | ● [12] | ✗ | ✓ |
| 2D Graphs | ● | ✓ | ✓ | ✓ | ✓ |
| 3D Graphs | ● [13] | ✓ | ✓ | ✓ | ✓ |
| ACE | ✓ | ✓ | ✓ | ✓ | ✗ |
| SOAP/Bartók | ✓ | ✓ | ✓ | ✓ | ✗ |
| Behler-Parrinello | ✓ | ✓ | ✓ | ✓ | ✗ |

---

[9]Dependent on canonicalization.

[10]Not invariant under changes in atom indexing.

[11]Each atom symbol is semantically unique.

[12]Requires post-processing for bijectivity.

[13]Yes there are some which are $E(3)$ invariant.

Table 2: Method Comparison II

|  | Topologically Accurate | Consider long-range interactions | Chemistry/Physics informed |
|---|---|---|---|
| Polyatomic Complex | ✓ | ✓ | ✓ |
| SMILES | ✗ | ✗ | 🟡 |
| SELFIES | ✗ | ✗ | 🟡 |
| 2D Graphs | ✗ | ✗ | 🟡 |
| 3D Graphs | 🟡 | ✗ | 🟡 |
| ACE | ✓ | ✓ | ✓ |
| SOAP/Bartók | ✓ | ✓ | ✓ |
| Behler-Parrinello | 🟡 | ✗ | ✓ |

## A.18 DATA: PHOTOSWITCHES TABLES

Table 3: Photoswitches Benchmark

| Kernel | Representation | E isomer $n$-$\pi$* wavelength (nm) | | | DFT E isomer $n$-$\pi$* wavelength (nm) | | |
|---|---|---|---|---|---|---|---|
| | | RMSE | MAE | CRPS | RMSE | MAE | CRPS |
| Tanimoto | SELFIES | $8.9 \pm 0.2$ | $5.7 \pm 0.1$ | 6.4 | $7.3 \pm 0.2$ | $3.6 \pm 0.1$ | 2.6 |
| Tanimoto | SMILES | $8.6 \pm 0.2$ | $5.1 \pm 0.1$ | 6.4 | $6.5 \pm 0.2$ | $3.2 \pm 0.04$ | 2.5 |
| Tanimoto | ECFP | $7.8 \pm 0.2$ | $4.4 \pm 0.1$ | 6.3 | $6.2 \pm 0.2$ | $3.1 \pm 0.1$ | 2.5 |
| Tanimoto | Fast Complex | $13.8 \pm 0.3$ | $7.0 \pm 0.1$ | 6.9 | $8.0 \pm 0.2$ | $2.5 \pm 0.1$ | 2.5 |
| Tanimoto | Deep Complex | $13.8 \pm 0.3$ | $7.0 \pm 0.1$ | 7.0 | $8.0 \pm 0.2$ | $2.5 \pm 0.1$ | 2.5 |
| WL | Graphs | $7.8 \pm 0.2$ | $4.4 \pm 0.1$ | 6.2 | $6.2 \pm 0.2$ | $3.1 \pm 0.1$ | 2.5 |

Table 4: Photoswitches Benchmark

| Kernel | Representation | E isomer $\pi$-$\pi$* wavelength (nm) | | | DFT E isomer $\pi$-$\pi$* wavelength (nm) | | |
|---|---|---|---|---|---|---|---|
| | | RMSE | MAE | CRPS | RMSE | MAE | CRPS |
| Tanimoto | SELFIES | $31.8 \pm 0.7$ | $22.2 \pm 0.5$ | 35.7 | $26.1 \pm 0.5$ | $17.1 \pm 0.2$ | 15.1 |
| Tanimoto | SMILES | $30.8 \pm 0.6$ | $22.1 \pm 0.3$ | 35.9 | $22.9 \pm 0.5$ | $14.6 \pm 0.2$ | 14.6 |
| Tanimoto | ECFP | $30.5 \pm 0.6$ | $21.6 \pm 0.3$ | 36.4 | $22.9 \pm 0.5$ | $13.7 \pm 0.3$ | 14.6 |
| Tanimoto | Fast Complex | $64.4 \pm 0.7$ | $52.1 \pm 0.5$ | 52.1 | $32.0 \pm 0.5$ | $15.7 \pm 0.5$ | 15.7 |
| Tanimoto | Deep Complex | $64.4 \pm 0.7$ | $52.1 \pm 0.5$ | 52.1 | $32.0 \pm 0.5$ | $15.7 \pm 0.5$ | 15.7 |
| WL | Graphs | $28.1 \pm 0.8$ | $17.9 \pm 0.4$ | 35.5 | $23.8 \pm 0.6$ | $15.3 \pm 0.3$ | 14.9 |

Table 5: Photoswitches Benchmark

| Kernel | Representation | Z isomer n-$\pi$* wavelength (nm) | | | DFT Z isomer n-$\pi$* wavelength (nm) | | |
|---|---|---|---|---|---|---|---|
| | | RMSE | MAE | CRPS | RMSE | MAE | CRPS |
| Tanimoto | SELFIES | $7.8 \pm 0.2$ | $3.9 \pm 0.1$ | 3.3 | $13.0 \pm 0.8$ | $4.1 \pm 0.1$ | 2.8 |
| Tanimoto | SMILES | $7.4 \pm 0.2$ | $3.4 \pm 0.1$ | 3.3 | $12.8 \pm 1.0$ | $3.2 \pm 0.2$ | 2.7 |
| Tanimoto | ECFP | $6.7 \pm 0.2$ | $3.0 \pm 0.1$ | 3.2 | $12.4 \pm 0.9$ | $4.0 \pm 0.1$ | 2.8 |
| Tanimoto | Fast Complex | $8.1 \pm 0.2$ | $3.6 \pm 0.1$ | 3.6 | $12.8 \pm 1.0$ | $2.8 \pm 0.3$ | 2.8 |
| Tanimoto | Deep Complex | $8.1 \pm 0.2$ | $3.6 \pm 0.1$ | 3.6 | $12.8 \pm 1.0$ | $2.8 \pm 0.3$ | 2.8 |
| WL | Graphs | $7.3 \pm 0.2$ | $3.1 \pm 0.1$ | 3.2 | $12.8 \pm 1.0$ | $2.8 \pm 0.3$ | 2.8 |

Table 6: Photoswitches Benchmark

| Kernel | Representation | Z Photostationary State (% isomer) | | | E Photostationary State (% isomer) | | |
|---|---|---|---|---|---|---|---|
| | | RMSE | MAE | CRPS | RMSE | MAE | CRPS |
| Tanimoto | SELFIES | $5.0 \pm 0.2$ | $2.3 \pm 0.1$ | 1.8 | $6.2 \pm 0.3$ | $2.8 \pm 0.1$ | 2.0 |
| Tanimoto | SMILES | $4.8 \pm 0.2$ | $2.2 \pm 0.1$ | 1.8 | $6.1 \pm 0.3$ | $2.6 \pm 0.1$ | 2.0 |
| Tanimoto | ECFP | $4.6 \pm 0.2$ | $2.0 \pm 0.1$ | 1.7 | $5.8 \pm 0.3$ | $2.5 \pm 0.1$ | 1.9 |
| Tanimoto | Fast Complex | $5.1 \pm 0.2$ | $1.8 \pm 0.1$ | 1.8 | $6.1 \pm 0.3$ | $2.1 \pm 0.1$ | 2.0 |
| Tanimoto | Deep Complex | $5.1 \pm 0.2$ | $1.8 \pm 0.1$ | 1.8 | $6.1 \pm 0.3$ | $2.1 \pm 0.1$ | 2.1 |
| WL | Graphs | $4.9 \pm 0.2$ | $2.1 \pm 0.1$ | 1.8 | $6.5 \pm 0.2$ | $2.8 \pm 0.1$ | 2.0 |

### A.19 FORCE MODELS

As described by Lin et al. (2019), in the context of force fields, the typical potential energy function is as shown in equation (1) (Lin & MacKerell, 2019).

$$V_{total} = \sum_{i=1}^{N_{bond}} V_{bond} + \sum_{i=1}^{N_{angle}} V_{angle} + \sum_{i=1}^{N_{dihedral}} V_{dihedral} + \sum_{i=1}^{N_{nonbonded}} V_{Nb} = \sum_{bonds} k_r(r - r_{eq})^2$$

$$+ \sum_{angles} k_\theta(\theta - \theta_{eq})^2 + \sum_{dihedrals} k_t[1 + \cos(n\omega - \gamma)] + \sum_{i<j} \left[ \frac{A_{ij}}{r_{ij}^{12}} - \frac{B_{ij}}{r_{ij}^6} \right] + \sum_{i<j} \frac{q_i q_j}{4\pi\epsilon r_{ij}} \quad (1)$$

For polyatomic complexes one can write the above equations as sums over incident cells. The definitions of the relations $\prec$, $\not\prec$, $\sim$ and $\not\sim$ are found in the Appendix A.9. For the $j$-th proton of dimension $\tau_k$ we let $\mathcal{N}_{PD_j^{\tau_k}} := \{PD_{j'}^{\tau_{k-1}} \mid PD_{j'}^{\tau_{k-1}} \prec PD_j^{\tau_k} \wedge (PD_{j'}^{\tau_{k-1}} \sim PD_j^{\tau_k} \vee PD_{j'}^{\tau_{k-1}} \not\sim PD_j^{\tau_k})\}$ contain all protons incident to proton $PD_j^{\tau_k}$. For the $j$-th neutron of dimension $\tau_k$ we similarly let $\mathcal{N}_{ND_j^{\tau_k}} := \{ND_{j'}^{\tau_{k-1}} \mid ND_{j'}^{\tau_{k-1}} \prec ND_j^{\tau_k} \wedge (ND_{j'}^{\tau_{k-1}} \sim ND_j^{\tau_k} \vee ND_{j'}^{\tau_{k-1}} \not\sim ND_j^{\tau_k})\}$ contain all neutrons incident to neutron $ND_j^{\tau_k}$. Finally for the $j$-th electron of dimension $\tau_k$ we let $\mathcal{N}_{ee_j^{\tau_k}} := \{ee_{j'}^{\tau_{k-1}} \mid ee_{j'}^{\tau_{k-1}} \prec ee_j^{\tau_k} \wedge (ee_{j'}^{\tau_{k-1}} \sim ee_j^{\tau_k} \vee ee_{j'}^{\tau_{k-1}} \not\sim ee_j^{\tau_k})\}$ contain all electrons incident to $ee_j^{\tau_k}$. For the $i$-th atom of dimension $k$ we similarly let $\mathcal{N}_{A_j^k} := \{A_{j'}^{k-1} \mid A_{j'}^{k-1} \prec A_j^k \wedge (A_{j'}^{k-1} \sim A_j^k \vee A_{j'}^{k-1} \not\sim A_j^k)\}$ contain all atoms incident to $A_j^k$. The set $\mathcal{N}_{A_j^k}$ then describes an atom's local environment. One can choose radius $r > 0$, and distance function $d$ such that:

$$\mathbf{ENV}_{A_\ell^k} := \{A_i^{k-1} \mid \forall(i, j) \, d(A_i^{k-1}, A_j^{k-1}) \le r \wedge A_i^{k-1} \prec A_\ell^k \wedge A_j^{k-1} \prec A_\ell^k\}. \quad (2)$$

$\mathbf{ENV}_{A_\ell^k}$ more generally defines environments in a molecule or polyatomic system.

This setup enables us to write out an equation analogous to (1). In particular, let $X$ be a polyatomic complex. Let $d_{n,k} : A_{i_1}^k \times A_{i_2}^k \times \cdots \times A_{i_n}^k \to \mathbb{R}$ be a valid metric defined on products of atoms (e.g. sup metric). Then let $\mathcal{V}_{bond} := \{(A_i^k, A_j^k) \mid d_{2,k}(A_i^k, A_j^k) < r_{bond} \, \forall k \le \dim(X)\}$. In essence, $\mathcal{V}_{bond}$ is the set of all pairs of atoms whose separation is only 1 bond, given by $r_{bond}$. Let $\mathcal{V}_{angle} := \{(A_i^k, A_j^k, A_\ell^k) \mid d_{3,k}(A_i^k, A_j^k, A_\ell^k) < r_{angle} \, \forall k \le \dim(X)\}$. In essence, $\mathcal{V}_{angle}$ is the set of all triples of atoms whose separation is only 2 bonds, thereby forming an angle, given by $r_{angle}$. Let $\mathcal{V}_{dihedral} := \{(A_i^k, A_j^k, A_\ell^k, A_m^k) \mid d_{4,k}(A_i^k, A_j^k, A_\ell^k, A_m^k) < r_{dih} \, \forall k \le \dim(X)\}$. In essence, $\mathcal{V}_{dihedral}$ is the set of all quadruples of atoms whose separation is only 3 bonds, thereby forming a dihedral potential or torsion potential, given by $r_{dih}$. Let $\mathcal{V}_{nb} := \{(A_i^k \times A_j^k) \mid d_{2,k}(A_i^k \times A_j^k) > r_{nb} \, \forall k \le \dim(X)\}$. In essence, $\mathcal{V}_{nb}$ is the set of all tuples of atoms whose separation is greater than 3 bonds given by $r_{nb}$. The definitions of $\mathcal{V}_{bond}$, $\mathcal{V}_{angle}$, $\mathcal{V}_{dihedral}$, and $\mathcal{V}_{nb}$ are motivated by the parameterizations given by the General Amber Force Field (Wang et al., 2004; Lin & MacKerell, 2019). For classification one can apply numerous force fields such as GAFF (Grimme, 2014; Senftle et al., 2016; Wang et al., 2004). The function $\rho$ describes the separation between the atoms. The function $\vartheta$ determines the angle between atoms. The function $\xi$ determines angles formed by the planes defined by atoms.

$$V_{total} = \sum_{v_r \in \mathcal{V}_{bond}} k_r(\rho(v_r) - r_{eq})^2 + \sum_{v_a \in \mathcal{V}_{angle}} k_\theta(\vartheta(v_a) - \theta_{eq})^2$$

$$+ \sum_{v_d \in \mathcal{V}_{dihedral}} k_t[1 + \cos(n \cdot \xi(v_d) - \gamma)] + \sum_{v_b \in \mathcal{V}_{nb}} \left[ \frac{A_{ij}}{\rho(v_b)^{12}} - \frac{B_{ij}}{\rho(v_b)^6} \right] + \frac{q_i q_j}{4\pi\epsilon\rho(v_b)} \quad (3)$$

More complex interactions can be modeled by considering sets like

$$\mathbf{Int}_n := \{(A_{i_1}^k \times \cdots \times A_{i_n}^k) \mid d_{n,k}(A_{i_1}^k \times \cdots \times A_{i_n}^k) < r \, \forall k \le \dim(X)\} \quad (4)$$

Note that all parameters can be chosen to reproduce experimental data or quantum mechanical calculations (Grimme, 2014; Wang et al., 2004). If one wishes to consider more complex dynamics, in the context of polyatomic complexes, doing so is possible. Suppose we wish to model the dynamics of evolution of a non-relativistic quantum system determined by the time-dependent Schrödinger equation (TDSE) (Scully, 2008). The time-dependent Schrödinger equation is defined as

$$i\hbar \frac{\partial}{\partial t}|\Psi(r, R, t)\rangle = \widehat{H}(r, R)|\Psi(r, R, t)\rangle \quad (5)$$

where $R = \{R_I\}$ are nuclear positions and $\{r_i\}$ are electronic positions. One can write approximations to the TDSE using polyatomic complexes. In physics one usually writes out the Hamiltonian as a sum of kinetic energy, potential energy, nuclear repulsion, electronic repulsion and electron-nuclear interaction (Jecko, 2014; Tsai et al., 2024).

$$\widehat{H}(r, R) = \widehat{T} + \widehat{V} = \widehat{T}_N + \widehat{T}_e + \widehat{V}_{NN} + \widehat{V}_{ee} + \widehat{V}_{eN} \tag{6}$$

We approximate this quantity in the following way.

Let $\mathbf{NUC} := \bigcup_{k=1}^{\dim(X)} \bigcup_{j=1}^{n_k} \left( \mathcal{N}_{PD_j^{\tau_k}} \cup \mathcal{N}_{ND_j^{\tau_k}} \right)$ and $\mathbf{ELEC} := \bigcup_{k=1}^{\dim(X)} \bigcup_{j=1}^{n_k} \mathcal{N}_{ee_j^{\tau_k}}$.

$$\widehat{H}(r, R) \approx - \sum_{I \in \mathbf{NUC}} \frac{-\hbar^2}{2M_I} \Delta_I^2 - \sum_{i \in \mathbf{ELEC}} \frac{-\hbar^2}{2m_e} \Delta_i^2 + \sum_{I \neq J \in \mathbf{NUC}} \frac{Z_A Z_B e^2}{4\pi\varepsilon_0 |\mathrm{pos}(I) - \mathrm{pos}(J)|}$$

$$+ \sum_{i \neq j \in \mathbf{ELEC}} \frac{e^2}{4\pi\varepsilon_0 |\mathrm{pos}(i) - \mathrm{pos}(j)|} - \sum_{i \in \mathbf{ELEC}, I \in \mathbf{NUC}} \frac{Z_A e^2}{4\pi\varepsilon_0 |\mathrm{pos}(I) - \mathrm{pos}(i)|} \tag{7}$$

where $\mathrm{pos}$ determines the nuclear or electronic positions and $\Delta_k$ is the Hodge-Laplacian as described in Appendix A.10. The Hodge-Laplacian terms approximate the derivatives with respect to the nuclear coordinates and electronic coordinates $\Delta_I^2 \approx \frac{\partial^2}{\partial \mathrm{pos}(I)^2}$ and $\Delta_i^2 \approx \frac{\partial^2}{\partial \mathrm{pos}(i)^2}$. The last four terms can be combined into an approximation for the electronic Hamiltonian $\widehat{H}_e(r, R) \approx \widetilde{H}_{\mathbf{NUC},\mathbf{ELEC}}$ (Albareda et al., 2021). Which enables us to write:

$$\widehat{H}(r, R) \approx - \sum_{I \in \mathbf{NUC}} \frac{-\hbar^2}{2M_I} \Delta_I^2 + \widetilde{H}_{\mathbf{NUC},\mathbf{ELEC}} \tag{8}$$

Then by substitution of (8) into (5) we see

$$i\hbar \frac{\partial}{\partial t} |\Psi(r, R, t)\rangle \approx \left( - \sum_{I \in \mathbf{NUC}} \frac{-\hbar^2}{2M_I} \Delta_I^2 + \widetilde{H}_{\mathbf{NUC},\mathbf{ELEC}} \right) |\Psi(r, R, t)\rangle \tag{9}$$

By the Born-Huang approximation we can approximate the total wave-function (Tsai et al., 2024).

$$|\Psi(r, R, t)\rangle = \sum_{\ell=0}^{\infty} |\chi_\ell(r; R)\Omega_\ell(R, t)\rangle \approx \sum_{l=0}^{\infty} |\phi_l \psi_l\rangle \tag{10}$$

Here, the $\phi_l$ correspond to the attaching maps $(\phi_{p,i})_{i \in I_p}$ and $(\phi_{n,i})_{i \in I_n}$ as in Definition 2.4. Similarly, $\psi_l$ correspond to the attaching maps $(\phi_{e,i})_{i \in I_e}$ as in Definition 2.4. Usually, $\chi_\ell(r; R)$ are the time-dependent nuclear wave-functions and $\Omega_\ell(R, t)$ are the solutions of the time independent SE.

## A.20 THE STRUCTURE Ato

We now introduce the idea of a monad of atomic complexes. Fundamentally, in probability theory, one can consider the category of topological spaces Top and equip it with a monad whose functor assigns a space $PA$ of probability measures on $X$ to each element $A$. In the case of Ato, a monad, we define a functor that assigns atomic complexes, elements of Top, to a space of probability measures.

**Definition A.19.** Ato is a monad whose objects are atomic complexes. Atoms are contained within $ob(\mathrm{Top})$. The morphisms are measurable functions encoding the probability that an atom forms a chemical bond. More explicitly:

Let $A_1$ and $A_2$ be atomic complexes and $f : A_1 \to A_2$ be a measurable function. Let $f_*$ denote the pushforward, $\eta$ be a natural transformation, $T$ be a functor, and $\mu : A \to TA$. Then the diagrams commute:

$$
\begin{array}{ccc}
A_1 \xrightarrow{f} A_2 & \quad TA \xrightarrow{T\eta} TTA & \quad TA \xrightarrow{\eta T} TTA & \quad TTTA \xrightarrow{T\mu} TTA \\
\downarrow{\mu} \quad \downarrow{\mu} & \quad \searrow \quad \downarrow{\mu} & \quad \searrow \quad \downarrow{\mu} & \quad \downarrow{\mu T} \quad \downarrow{\mu} \\
TA_1 \xrightarrow{f_*} TA_2 & \quad TA & \quad TA & \quad TTA \xrightarrow{\mu} TA
\end{array}
$$

*Remark* A.20. The full proof is found below in this section. At a high level, since atomic complexes are topological spaces, they can be equipped with a $\sigma$-algebra, and the morphisms are measurable; the result follows mutandis mutatis from the Giry monad on Meas (Giry, 2006).

*Remark* A.21. This property allows a user to encode and make statements about the probability of two atomic complexes forming a chemical bond in a compact way. In essence, the $\frac{1}{r}$ Coulomb potential that gives the attraction between charges.

*Proof.* We want to show that $\mathrm{Ato}$ defines a monad structure. Since atomic complexes are topological spaces that you can equip with a $\sigma$-algebra and the morphisms are measurable the result follows mutandis mutatis from the Giry monad on $\mathrm{Meas}$ Giry (2006). You can generate the $\sigma$-algebra by the integration functions $\varepsilon_f : PA \to \mathbb{R}$ by

$$p \mapsto \int f dp,$$

for all $f : A \to [0,1]$ measurable. You can define multiplication in the usual way:

$$E_\pi(X) := \int_{PA} p(X) d\pi(p)$$

when given $A$ and measure $\pi \in PPA$. By composition of morphisms you can reproduce the Chapman-Kolmogorov equation for general measures given $f : A_1 \to PA_2$ and $g : A_2 \to PA_3$ we see:

$$(f \circ g)(a_1)(Z) = \int_{A_2} g_{a_2}(Z) df_{a_1}(a_2)$$

for each $a_1 \in A_1$ and $Z \subseteq A_3$. This composition is associative and unital. Thus we have a monad structure. $\square$

## A.21 COMPUTE COST

All reported experiments can be reproduced using an AWS m7g.4xlarge instance (16 vCPU, 64 GiB) in under three hours per experiment. The full project required more compute than the experiments reported in the paper. This is because we conducted a wider variety of experiments on different datasets which are not reported. Certain experiments require resources equivalent to that of an AWS p5.48xlarge (192 vCPU, 2TiB, 8 GPU-H100) and approximately ten hours per experiment.

