# OpenReview forum: "POLYATOMIC COMPLEXES: A TOPOLOGICALLY INFORMED LEARNING REPRESENTATION FOR ATOMISTIC SYSTEMS"
_ICLR.cc/2025/Conference — Submitted to ICLR 2025_

### Official Review · Reviewer_h22M · 2024-10-18

**Soundness:** 2
**Presentation:** 1
**Contribution:** 2
**Rating:** 3
**Confidence:** 4

**Summary:**

This paper introduces a new representation for chemical structures, designed to offer a more robust alternative to existing methods such as SMILES and fingerprints. By incorporating the physical characteristics of the target material, the proposed representation seems to demonstrate several desirable features that are critical for an effective chemical representation.

**Strengths:**

S1. The author asserts that the proposed representation satisfies several key criteria, including invariance, uniqueness, continuity and differentiability, generalizability, efficiency, topological accuracy, long-range interaction handling, and incorporation of chemical and physical insights.

S2. These claims are supported, at least partially, by mathematical discussions.

**Weaknesses:**

The paper appears to be in a draft stage, possibly intended to gather feedback before submitting to another conference or journal. This is inferred from several issues:

W1: The overall structure, figures, and explanations appear incomplete. For instance, the contents and sentences in Figure 1 are too small and difficult to understand. Section 2 lacks sufficient explanations connecting the lemmas and theorems, making it challenging to follow the argument. The description of the algorithm in Section 2.3 is brief and does not fully clarify the logic. Additionally, Figure 2 is uninformative and takes up space that could be better used to elaborate on the explanation and discussion.

W2: The experiments do not convincingly demonstrate the effectiveness or utility of the new representation. Across most datasets, the performance of the proposed representation is suboptimal, failing to justify its usefulness. The experiments focus solely on property prediction performance. It would be more informative to include comparisons of computational cost, assessments of generalizability, evaluations of long-range interaction capabilities against machine learning interatomic potential (MLIP) models, and tests on datasets where other representations struggle due to issues such as lack of uniqueness. In such cases, MLIP models could serve as a meaningful baseline due to their ability to represent diverse materials.

W3:  Due to the suboptimal layout, the sections on experiments and discussions (Sections 3 and 4) are very brief and lack insightful analysis that could demonstrate the effectiveness of the new representation.

In conclusion, the current draft requires substantial revisions before it can be considered for acceptance. It may be more appropriate to submit to a journal focused on computational chemistry, where an unlimited number of pages would be allowed for a more thorough explanation of the new representation and its advantages.

**Questions:**

W2. What new tasks could be uniquely enabled by the proposed representation, and how do the authors plan to validate its performance in these tasks?

For additional concerns, please refer to the "Weaknesses" section above.

---

> ### Author Response · Authors · 2024-11-26
>
> We would like to thank the reviewer for their insightful feedback. We address each weakness point by point.
>
> 1. The overall structure, figures, and explanations appear incomplete. For instance, the contents and sentences in Figure 1 are too small and difficult to understand. Section 2 lacks sufficient explanations connecting the lemmas and theorems, making it challenging to follow the argument. The description of the algorithm in Section 2.3 is brief and does not fully clarify the logic. Additionally, Figure 2 is uninformative and takes up space that could be better used to elaborate on the explanation and discussion.
>
> We have made the following changes:
> - We move the algorithm to the appendix to ensure better flow throughout the paper.
> - We remove figure 2.
> - We systematically reduce whitespace and extra spaces for readability and formatting.
> - We introduce a new paragraph in the introduction clearly explaining the Graphical Abstract (Figure 1).
> In this paragraph we delineate what the representation corresponds to and its final form as a tensor.  Moreover in section 2.3 we better explain how the representation used in the experiments is algorithmically constructed.
>
> 2. The experiments do not convincingly demonstrate the effectiveness or utility of the new representation. Across most datasets, the performance of the proposed representation is suboptimal, failing to justify its usefulness. The experiments focus solely on property prediction performance. It would be more informative to include comparisons of computational cost, assessments of generalizability, evaluations of long-range interaction capabilities against machine learning interatomic potential (MLIP) models, and tests on datasets where other representations struggle due to issues such as lack of uniqueness. In such cases, MLIP models could serve as a meaningful baseline due to their ability to represent diverse materials.
>
> We agree with this point. It would be informative to include comparisons of computational cost, assessments of generalizability, and evaluations of long-range interaction capabilities. We provide a table in the Appendix A.17 making this comparison. We can explain why the accuracy is limited for the datasets we chose. In essence
> - We want the comparison between our representation and other representations (SMILES, etc.) to be reasonably fair.
>
> To elaborate on the above bullet point, We essentially map the higher dimensional topological representation to a lower dimensional subspace which results in notably less geometric information being used by the model. This is done purely out of necessity because the model leverages the Tanimoto Kernel. The reason we choose the Tanimoto kernel is that a direct comparison can be made with SMILES, SELFIES etc. among other representations. This is what me mean by "fair comparison." In essence, we can use the same kernel for everything. However, using a kernel better suited to the representation, and avoiding the decision to map to a lower dimensional space, would very likely improve RMSE.
>
> There are datasets wherein a lack of uniqueness (by SMILES etc.) would cause severely degraded performance but would a comparison be fair. Couldn't one argue that SMILES wasn't developed for such purposes. Furthermore, we could compare to MLIP's but would it be fair to compare MLIP's to SMILES etc. This kind of illustrates the first point.
>
> 3. Due to the suboptimal layout, the sections on experiments and discussions (Sections 3 and 4) are very brief and lack insightful analysis that could demonstrate the effectiveness of the new representation.
>
> This is a fair point. We restructured the paper so that the organization is improved. We moved the algorithms to the appendix and added more information to figure 1 in a paragraph entitled "Graphical Abstract." We also removed figure 2. We agree that the analysis could be deeper.
>
>
> Q1. (W2) What new tasks could be uniquely enabled by the proposed representation, and how do the authors plan to validate its performance in these tasks?
>
> Excellent question! We did develop this representation with particular future applications in mind. Some of which aren't unique. Here is a non-comprehensive list:
>
> - Determining dipole moment
> - Predicting solid phase heat of formation
> - Predicting HOMO–LUMO gap
> - Predicting spin-state ordering, spin-state energetics, Spin-state splittings
> - Analyzing Hartree–Fock exchange
> - Chapman–Jouguet (C–J) detonation velocity, detonation pressure, and temperature
> - First-principles geometry optimizations (DFT gas-phase geometry)
> - Extrapolating pure exchange–correlation functionals to hybrids
> - Determining total electronic energy
>
> We are also exploring the intersection of force fields and our representation please see appendix A.19 for a more in depth discussion. We are also experimenting with diffuse functions / basis set and orbital geometry augmentations.

---

> ### Comment · Reviewer_h22M · 2024-11-27
> **reply**
>
> Thank you for the author's serious consideration of my comment.
>
> Although the comment seems interesting, it would be a little unclear where have been changed. Please consider to change font color to the modified part.
>
> Maybe I can modify my present very bad evaluation score into a moderate one after checking it.

---

> ### Author Response · Authors · 2024-11-27
> **re: reply**
>
> The major writing changes are now in blue.
>
> We can't put any deletions in blue but figure 2 and the whitespace are removed.
>
> The table comparing approaches is in Appendix A.17.
>
> We also added bolding where appropriate in the tables.
>
> We thank the reviewer for their consideration.

---

> ### Comment · Reviewer_h22M · 2024-11-28
>
> Thank you for the changing the color of the updated part.
> Now it is much better than the previous one but I still recommend to polish the following part:
>
> 1. Figure 1's font is still too small and not readable:
>
> As the other reviewers pointed out, Figure 1's font is still too small and would be quite difficult to read, in particular, when printing in A4 paper.
>
> 2. Seemingly, the main part's experiment is not refreshed, which is one of the most concerning part, as described previously. They are still showing that the proposed method provides only comparable or smaller information in comparison to the existing methods, which is partly due to the dimension reduction for using kernel method, as the authors pointed out.
> As mentioned previously, the experiment method should show the effectiveness of the proposed method, which is still failed in the previous version. The authors should find a more appropriate experiment to show the new method is useful.
> In addition, the discussion in terms of the experiments are too short and shallow. Maybe some of the experiments can be moved into appendix and tables can be merged using subfigure or something. The saved space can be used further analysis or new experiments.
> I think the authors can learn a lot of idea from SELFIES's paper [1] to validate the effectiveness of the newly proposed chemical expression.
>
> [1] DOI 10.1088/2632-2153/aba947
>
> Anyway, I slightly modify the previous evaluation but it still needs major update.

---

### Official Review · Reviewer_wA48 · 2024-10-27

**Soundness:** 3
**Presentation:** 3
**Contribution:** 2
**Rating:** 5
**Confidence:** 4

**Summary:**

This paper  developed a new molecular representation encoding atomistic systems called polyatomic complexes, which constructs mathematical models by CW complexes and connects different elementary particles through gluing maps.

Besides, this paper expanded the criteria for good molecular representations and demonstrated that the proposed method satisfies all these constrains.

The authors also demonstrated the performance of this representation on several benchmark datasets in chemistry and materials science.

**Strengths:**

This paper demonstrates strong originality at the theoretical level, introducing a method grounded in a rigorous mathematical model that meets stringent constraints.

**Weaknesses:**

The method presented in this paper exhibits weak predictive accuracy across multiple benchmark datasets.
Although the polyatomic complexes method characterizes at the electronic level, its performance is still less than that of the molecular level methods, e.g. SMILES and SELFIES.

**Questions:**

- Time cost of computing representations compared with other methods is not provided to evaluate the **efficiency** constrain.
- The theory behind the model is excellent, but the practical implementation employs many simplifications and approximations to save computational efficiency, raising concerns that it may not achieve the capabilities claimed in theory.

- We notice that the authors have added more stringent constraints of molecular representations based on [1], such as Topological Accuracy, Long-range interactions and Chemical and Physical Informedness. We are skeptical about the necessity of these additional criteria.  Molecular representation is a lossy compression of information, and expecting a perfect representation of all properties may be challenging. Perhaps focusing on certain aspects while neglecting others could be more helpful for specific tasks. For example, electronic structure information and topological accuracy may not be very helpful in predicting macroscopic properties such as solubility, which is also experimentally demonstrated in Table2 and Table 3. Have the authors discussed why these criteria are added and are these additional criteria really useful?



[1] M. F. Langer, A. Goeßmann, and M. Rupp, “Representations of molecules and materials for interpolation of quantum-mechanical simulations via machine learning,” npj Comput Mater, vol. 8, no. 1, p. 41, Mar. 2022, doi: 10.1038/s41524-022-00721-x.

---

> ### Author Response · Authors · 2024-11-26
> **Re: Review of Submission3311 by Reviewer wA48**
>
> We would like to thank the reviewer for their insightful feedback. We address each weakness point by point.
>
> 1. The method presented in this paper exhibits weak predictive accuracy across multiple benchmark datasets. Although the polyatomic complexes method characterizes at the electronic level, its performance is still less than that of the molecular level methods, e.g. SMILES and SELFIES.
>
> We agree. We describe this as a weakness and it is important to clarify how improvements can be made and the underlying cause. There are numerous improvements one can make and ways in which accuracy could improve. We discuss them below.
>
> The first improvement one can make is including a representation of orbitals (s/p/d/f) and orbital geometry. Additionally, we empirically observe that the Tanimoto kernel is poorly suited to the representation developed. Using a better kernel would likely significantly improve accuracy with relatively minor impact to cost. Moreover, leveraging more of the features we provide in our representation, in our learning method, would likely improve accuracy substantially. There are two reasons why we do not leverage all of the features provided by our representation in the model. We discuss this briefly in our discussion.
>
> - We want the comparison between our representation and other representations (SMILES, etc.) to be reasonably fair. In essence we only train models with reduced geometric features.
> - We wanted the computational cost of our experiments to be low.
>
> To elaborate on the first bullet point, We essentially map the higher dimensional topological representation to a lower dimensional subspace which results in notably less geometric information being used by the model. This is done purely out of necessity because the model leverages the Tanimoto Kernel. The reason we choose the Tanimoto kernel is that a direct comparison can be made with SMILES, SELFIES etc. among other representations. This is what me mean by "fair comparison." In essence, we can use the same kernel for everything. However, using a kernel better suited to the representation would very likely improve RMSE. Developing such kernels is a rather new and active area of research.

---

> ### Author Response · Authors · 2024-11-26
> **Re: Questions by Reviewer wA48**
>
> 1. Time cost of computing representations compared with other methods is not provided to evaluate the efficiency constraint.
>
> We provide a proof of time complexity in Theorem 2.12 and Appendix A.15. We additionally added a table in the Appendix A.17 directly comparing all the methods including across the efficiency constraint. We agree that this should be clearer. Please see the revised manuscript for the table.
>
> 2. The theory behind the model is excellent, but the practical implementation employs many simplifications and approximations to save computational efficiency, raising concerns that it may not achieve the capabilities claimed in theory.
>
> This is a good point! We believe that the practical aspects can definitely be improved at the cost of increased time complexity. We additionally didn't include some features and setup our feature engineering so we could use the Tanimoto kernel. The reason for doing so is that we wanted the comparison to be fair with regard to the other representations. In essence, we wanted to use the same kernel for every representation. However, we empirically observe that the tanimoto kernel isn't well suited to our representation. We are quite sure that using a better kernel and including diffuse functions / basis set + orbital geometry will lead to significantly better accuracy.
>
> 3. We notice that the authors have added more stringent constraints of molecular representations based on [1], such as Topological Accuracy, Long-range interactions and Chemical and Physical Informedness. We are skeptical about the necessity of these additional criteria. Molecular representation is a lossy compression of information, and expecting a perfect representation of all properties may be challenging. Perhaps focusing on certain aspects while neglecting others could be more helpful for specific tasks. For example, electronic structure information and topological accuracy may not be very helpful in predicting macroscopic properties such as solubility, which is also experimentally demonstrated in Table2 and Table 3. Have the authors discussed why these criteria are added and are these additional criteria really useful?
>
> Excellent question! Our representation provides enough flexibility such that one can choose to exclude or include certain features. The code is written in a modular enough way such that one can make those changes. The primary reasoning for why we include these constraints is that we wanted our theoretical model to be as stringent as possible. In essence, we were aiming for a representation which is, in theory, as perfect as possible and demonstrate that such an implementation was plausible. Following the constraints from Langer (which we believe are minimal for ensuring a representation is representative and effective) and showing that we satisfy additional constrains strengthens the use of our representation. As it leaves open applications wherein electronic structure is relevant. In essence our representation is simultaneously reasonably suited to tasks wherein macroscopic properties are important as well as where electronic structure is important. There are use cases where electronic structure is applicable. We developed this representation with other tasks in mind as well.

---

### Official Review · Reviewer_xn4r · 2024-10-31

**Soundness:** 2
**Presentation:** 2
**Contribution:** 2
**Rating:** 3
**Confidence:** 3

**Summary:**

This paper proposes a method using CW-complex to build representations for atomistic systems. It consists two stages: in the first stage the CW complex representation is built for single atom based on proton, neutron and electrons; in the second stage, the CW complex representation is built for multiple atoms using the representations from single atoms. Numerical experiments have been performed on multiple benchmarks and compared with other baselines.

**Strengths:**

- The idea of building representation from topological information is novel.
- Numerical experiments have been done and compared comprehensively

**Weaknesses:**

- Some statements related to relevant works have factual incorrectness. For example,
    - In line 142-146, in the description of ACE, the computational cost is not N! for permutations, and (J N) for J-neighbors. I suggest the authors to check the original ACE paper (Ralf Drautz, PRB 2019). The permutation invariant within atoms of the same element is achieved by simply summing over basis function of neighbor displacement, and by using clusters it also circumvents the (J N) summation.
    - In line 391, the authors state that the electromagnetic force, strong nuclear force and weak nuclear force hold the atom together, which is not a physically correct statement.
- The construction with proton, neutron, and electron is not very physically meaningful for the tasks of interests. This is just to obtain a representation for a single atom, which basically includes only the element and isotope information. A direct embedding on the element type will be simpler and achieve the same goal and learnable.
- In the comparison with other methods, this method does not show impressive results. E.g. on benchmarks like ESOL, FreeSolv, ChEMBL, the error is much worse than SMILES and SELFIES

**Questions:**

It is not very clear to me how the CW complex is implemented. The authors should clarify the form of the final representation (e.g. scalar/vector/tensor/graph, what is the dimension) which is input to the regression model.

---

> ### Author Response · Authors · 2024-11-26
> **Re: Review of Submission3311 by Reviewer xn4r**
>
> We would like to thank the reviewer for their insightful feedback. We address each weakness point by point.
>
> 1. Some statements related to relevant works have factual incorrectness. For example, In line 142-146, in the description of ACE, the computational cost is not N! for permutations, and (J N) for J-neighbors. I suggest the authors to check the original ACE paper (Ralf Drautz, PRB 2019). The permutation invariant within atoms of the same element is achieved by simply summing over basis function of neighbor displacement, and by using clusters it also circumvents the (J N) summation. In line 391, the authors state that the electromagnetic force, strong nuclear force and weak nuclear force hold the atom together, which is not a physically correct statement.
>
> We absolutely agree and thank the reviewer for pointing out this oversight! We make the corrections in the revised manuscript. We made a mistake and apologize for the factual inaccuracies on lines 142-146 and 391. These statements have been corrected and we provided the correct citations. Thank you!
>
> 2. The construction with proton, neutron, and electron is not very physically meaningful for the tasks of interests. This is just to obtain a representation for a single atom, which basically includes only the element and isotope information. A direct embedding on the element type will be simpler and achieve the same goal and learnable.
>
> This is a good point! We respectfully disagree with part of the point made. The primary issue with directly embedding the element type is none of the important geometric/topological information at the electronic structure level is included. The electronic structure is intrinsically linked to chemical bonding and by extension the topology of the molecule. There are intended use cases of this representation wherein we include the orbital geometry, orbitals, the basis set and diffuse functions. We do agree however, that the tasks within our experiments section would not find these features very physically meaningful. However, this is a good point in the context of the experiments. We include this information mostly with future applications in mind.
>
> 3. In the comparison with other methods, this method does not show impressive results. E.g. on benchmarks like ESOL, FreeSolv, ChEMBL, the error is much worse than SMILES and SELFIES.
>
> We agree. We describe this as a weakness and it is important to clarify how improvements can be made and the underlying cause. There are numerous improvements one can make and ways in which accuracy could improve. We discuss them below.
>
> The first improvement one can make is including a representation of orbitals (s/p/d/f) and orbital geometry. Additionally, we empirically observe that the Tanimoto kernel is poorly suited to the representation developed. Using a better kernel would likely significantly improve accuracy with relatively minor impact to cost. Moreover, leveraging more of the features we provide in our representation, in our learning method, would likely improve accuracy substantially. There are two reasons why we do not leverage all of the features provided by our representation in the model. We discuss this briefly in our discussion.
>
> - We want the comparison between our representation and other representations (SMILES, etc.) to be reasonably fair. In essence we only train models with reduced geometric features.
> - We wanted the computational cost of our experiments to be low.
>
> To elaborate on the first bullet point, We essentially map the higher dimensional topological representation to a lower dimensional subspace which results in notably less geometric information being used by the model. This is done purely out of necessity because the model leverages the Tanimoto Kernel. The reason we choose the Tanimoto kernel is that a direct comparison can be made with SMILES, SELFIES etc. among other representations. This is what me mean by "fair comparison." In essence, we can use the same kernel for everything. However, using a kernel better suited to the representation, and avoiding the decision to map to a lower dimensional space, would very likely improve RMSE. Developing such kernels is a rather new and active area of research.

---

> ### Author Response · Authors · 2024-11-26
> **Re: Questions by Reviewer xn4r**
>
> Q1. It is not very clear to me how the CW complex is implemented. The authors should clarify the form of the final representation (e.g. scalar/vector/tensor/graph, what is the dimension) which is input to the regression model?
>
> This is an excellent point. We describe this better in Section 2.3 and included a new paragraph entitled "Graphical Abstract" wherein we clarify how we construct the complex at a higher level and state that the final representation is a tensor.

---

### Official Review · Reviewer_4fsS · 2024-11-02

**Soundness:** 2
**Presentation:** 2
**Contribution:** 2
**Rating:** 5
**Confidence:** 3

**Summary:**

The paper proposes a novel way of representing molecules and materials by incorporating topological information through a series of cell complexes that satisfy all structural, geometric, efficiency, and generalizability.

**Strengths:**

- A novel way of constructing a representation for molecules satisfying key ingredients
- Incorporating structural, topological information to construct representation

**Weaknesses:**

- The paper's writing lacks clarity, making it difficult to follow.
- The latter part of the introduction, which describes and compares various representation approaches, could be more effectively presented in a table. This would also help clarify where the proposed method fits within current methods, with detailed explanations that can be moved to an appendix.
- Including a subsection on notation would improve readability by clearly defining the meaning of each symbol used.
- Although the representation meets vital criteria, it still underperforms in benchmarks (e.g., ESOL and FreeSolv). Could this be due to limitations in the current method's ability to leverage higher-level topological information from this representation, or are there other underlying reasons?

**Questions:**

- Could the authors discuss the time complexity of the proposed algorithm for constructing the complex, along with an ablation study? This would help clarify the potential trade-offs associated with the method.
- During the construction of the atom complex, is there a way to incorporate neutron and proton arrangements in energy levels (similar to the nuclear shell model)?
- Regarding the uniqueness of the representation, can the method distinguish between chiral molecules or stereoisomers of the same molecule by obtaining distinct representations?
- Given that the method represents an atom complex by decomposing electrons, neutrons, and protons and then reconstructing the molecule, how does it handle bond formation between electron-donating and electron-accepting atoms, where the atoms may carry a charge? or in general, representing a charged molecule?

---

> ### Author Response · Authors · 2024-11-26
> **Re: Official Review of Submission3311 by Reviewer 4fsS**
>
> We would like to thank the reviewer for their insightful feedback. We address each weakness point by point.
>
> 1. The paper's writing lacks clarity, making it difficult to follow.
>
> We wholeheartedly agree that the readability and organization of the paper could be improved. In order to address readability we make the following changes to the manuscript writing:
>
> - We move the algorithm to the appendix to ensure better flow throughout the paper.
> - We systematically reduce whitespace and extra spaces for readability and formatting.
> - We introduce a new paragraph in the introduction clearly explaining the Graphical Abstract (Figure 1).
>
> In this paragraph we delineate what the representation corresponds to and its final form as a tensor. Moreover in section 2.3 we better explain how the representation used in the experiments is algorithmically constructed.
>
> 2. The latter part of the introduction, which describes and compares various representation approaches, could be more effectively presented in a table. This would also help clarify where the proposed method fits within current methods, with detailed explanations that can be moved to an appendix.
>
> We agree. We produce this table in the appendix of the revised manuscript. Please see Appendix A.17 "METHOD COMPARISON." We view the discussion as part of the literature review, however. Do you think an effective compromise would be reducing the explanations and adding more detail to the appendix?
>
> 3. Including a subsection on notation would improve readability by clearly defining the meaning of each symbol used.
>
>  We agree that the notation could be clearer. Would providing a table in the appendix satisfy this requirement? Some of our notation is standard from algebraic topology and other things we define. Should we clarify things like the \hookrightarrow or symbols we define? In essence which parts of the notation are most unclear and what should we focus on defining? We definitely agree though that including descriptions of the notation would improve readability and clarity!
>
> 4. Although the representation meets vital criteria, it still underperforms in benchmarks (e.g., ESOL and FreeSolv). Could this be due to limitations in the current method's ability to leverage higher-level topological information from this representation, or are there other underlying reasons?
>
> Your intuition is precisely correct. The choice of kernel function (Tanimoto) kernel limits the models ability to leverage higher level topological information. This is one of the biggest reasons for the underperformance in some benchmarks. The reason we choose the Tanimoto kernel is that a direct comparison can be made with SMILES, SELFIES etc. among other representations. This is what me mean by "fair comparison." In essence, we can use the same kernel for everything. However, using a kernel better suited to the representation would very likely improve RMSE. Developing such kernels is a rather new and active area of research. This is briefly touched on in the Discussion.

---

> ### Author Response · Authors · 2024-11-26
> **Re: Questions by  Reviewer 4fsS**
>
> Q1. Could the authors discuss the time complexity of the proposed algorithm for constructing the complex, along with an ablation study? This would help clarify the potential trade-offs associated with the method.
>
> Good question! We provide a proof of the time complexity being $O(S)$ in the Appendix A.15 and mention it in Theorem 2.12. We wonder about the potential benefits of an ablation study for one simple reason. Essentially we primarily leverage geometric information in the model and exclude other important features. Doesn't this imply that there wouldn't be much to remove? We don't use any of the information related to forces, and don't augment with diffuse functions, orbitals or orbital geometry in the experiments. However, we can definitely include an ablation study if it improves clarity! What replacements or removals would you suggest?
>
> Q2. During the construction of the atom complex, is there a way to incorporate neutron and proton arrangements in energy levels (similar to the nuclear shell model)?
>
> Yes one could theoretically do this as an augmentation after the fact. Our representation is flexible enough to support this. We believe that including orbitals, orbital geometry and diffuse functions would likely be a good augmentation as well.
>
> Q3. Regarding the uniqueness of the representation, can the method distinguish between chiral molecules or stereoisomers of the same molecule by obtaining distinct representations?
>
> Good question! Yes. This is a requirement of uniqueness.
>
> Q4. Given that the method represents an atom complex by decomposing electrons, neutrons, and protons and then reconstructing the molecule, how does it handle bond formation between electron-donating and electron-accepting atoms, where the atoms may carry a charge? or in general, representing a charged molecule?
>
> Good question! We model electrons as spheres with fixed radius paired with their corresponding wavefunction. We account for bond formation by modeling the probability that any two atoms form a chemical bond (see Appendix A.20). This accounts for charge. One can further augment with orbitals, a basis set, and diffuse functions. Basis sets are sets of functions which are used to represent the electronic wave functions in the Hartree-Fock method or DFT. A common addition to the basis sets is the addition of diffuse functions. Diffuse basis functions can describe anions or dipole moments and can be used for the accurate modeling of intra and inter-molecular bonding.

---

### Official Review · Reviewer_J9iZ · 2024-11-04

**Soundness:** 3
**Presentation:** 2
**Contribution:** 2
**Rating:** 3
**Confidence:** 2

**Summary:**

This paper proposes a new representation (polyatomic complexes) for chemical systems, aimed at capturing topological properties of molecules. In particular, the representation 'stitches' together representations of different neutrons, protons and electrons. The authors prove that polyatomic complexes satisfy (rotational, translational, permutational) invariances, continuity (with respect to atomic positions), generalizability (in the sense of being well-defined for arbitrary chemical compounds) and topological accuracy (upto electronic structure).
The authors perform experiments on the photoswitches, ESOL, FreeSolv and Matbench JDFT2D dataset with two variants of their method (Fast Complex and Deep Complex), and find comparable performance to state-of-the-art methods.

My main concerns are the readability of the paper together with the lack of convincing empirical results (which the authors also admit to). It would be nice to come up with a task where this representation is provably better than existing methods.

**Strengths:**

- The idea of polyatomic complexes for describing chemical systems is novel as far as I know. I think there are some good ideas in this paper, and it would be nice to improve the writing to flesh them out a bit more.
- The choice of datasets and baselines is quite comprehensive and interesting. I really appreciate the authors' honesty here in showing all results even when their representation is not the best-performing.

**Weaknesses:**

- This paper is quite difficult to read (but this might also be my lack of familiarity with the notation). It is hard to understand what exactly the representation corresponds to. Some more figures describing the construction of polyatomic complexes could help.
- The formatting could be improved significantly. For example, Figure 1 is barely readable due to small text, and there is extra whitespace throughout the paper (eg. "( 2.12)" on page 3 and so forth.)
- The modeling assumption of an electron as a single sphere is quite chemically inaccurate (especially for delocalized systems). It would be good for the authors to discuss this weakness and how to improve this aspect of their representation.
- Experimental results are not super convincing, given that the RMSE of their method is often much higher than state-of-the-art methods. The authors do mention this as a weakness, but I would like to see how improvements can be made.

**Questions:**

- 'Generalizability': I felt that this property could be differently named, in order to avoid confusion with its definition in the ML literature. Perhaps, 'completeness' captures the essence of the property better?
- e3nn is not a chemical representation, it is a framework for training E(3)-equivariant neural networks (which may be used to learn representations of chemical systems).
- Section 3.1: "Since we believe the values to be missing at random, we utilize mean imputation, instead of discarding experimental data". This is a big assumption, do the results change significantly if missing values are removed?
- Please highlight your method in all tables for clarity.
- What exactly is the role of the random matrix denoting proton-electron interactions? Are these forces/energies learned?
- Some of the definitions seem arbitrary. eg. where does the constant 2.8 fm come from in Definition 2.1?

---

> ### Author Response · Authors · 2024-11-26
> **Re: Official Review of Submission3311 by Reviewer J9iZ**
>
> We would like to thank the reviewer for their insightful feedback. We address each weakness point by point.
>
> 1. This paper is quite difficult to read (but this might also be my lack of familiarity with the notation). It is hard to understand what exactly the representation corresponds to. Some more figures describing the construction of polyatomic complexes could help.
>
> We wholeheartedly agree that the readability of the paper could be improved. In order to address readability we make the following changes to the manuscript writing:
> - We move the algorithm to the appendix to ensure better flow throughout the paper.
> - We systematically reduce whitespace and extra spaces for readability and formatting.
> - We introduce a new paragraph in the introduction clearly explaining the Graphical Abstract (Figure 1). In this paragraph we delineate what the representation corresponds to and its final form as a tensor. Moreover in section 2.3 we better explain how the representation used in the experiments is algorithmically constructed.
>
> 2. The formatting could be improved significantly. For example, Figure 1 is barely readable due to small text, and there is extra whitespace throughout the paper (eg. "( 2.12)" on page 3 and so forth.)
>
> We agree that the formatting could be improved significantly. We eliminate Figure 2, move the algorithms to the appendix, and provide more explanation for Figure 1. We introduce a paragraph entitled "Graphical Abstract" in order to better explain the figure and overall representation. We systematically eliminate unnecessary whitespace.
>
> 3. The modeling assumption of an electron as a single sphere is quite chemically inaccurate (especially for delocalized systems). It would be good for the authors to discuss this weakness and how to improve this aspect of their representation.
>
> We agree that the assumption of an electron as a single sphere is chemically inaccurate. However, we represent an electron as a sphere with its corresponding wavefunction. We do believe that the reader may overlook this detail however, so we clean up the writing slightly in this area. Moreover, proposed extensions of the representation such as representing shells or orbitals and orbital geometry would address this further.
>
> 4. Experimental results are not super convincing, given that the RMSE of their method is often much higher than state-of-the-art methods. The authors do mention this as a weakness, but I would like to see how improvements can be made.
>
> We agree entirely. We describe this as a weakness and it is important to clarify how improvements can be made. There are numerous improvements one can make and ways in which accuracy could improve. We discuss them below.
>
> The first improvement one can make is including a representation of orbitals (s/p/d/f) and orbital geometry. Additionally, we empirically observe that the Tanimoto kernel is poorly suited to the representation developed. Using a better kernel would likely significantly improve accuracy
> with relatively minor impact to cost. Moreover, leveraging more of the features we provide in our
> representation, in our learning method, would likely improve accuracy substantially. There are two reasons why we do not leverage all of the features provided by our representation in the model.
> - We want the comparison between our representation and other representations (SMILES, etc.) to be reasonably fair. In essence we only train models with reduced geometric features.
> - We wanted the computational cost of our experiments to be low.
>
> To elaborate on the first bullet point, We essentially map the higher dimensional topological representation to a lower dimensional subspace which results in notably less geometric information being used by the model. This is done purely out of necessity because the model leverages the Tanimoto Kernel. The reason we choose the Tanimoto kernel is that a direct comparison can be made with SMILES, SELFIES etc. among other representations. This is what me mean by "fair comparison." In essence, we can use the same kernel for everything. However, using a kernel better suited to the representation, and avoiding the decision to map to a lower dimensional space, would very likely improve RMSE. Developing such kernels is a rather new and active area of research.

---

> ### Author Response · Authors · 2024-11-26
> **Re: Questions by Reviewer J9iZ**
>
> Q1. 'Generalizability': I felt that this property could be differently named, in order to avoid confusion with its definition in the ML literature. Perhaps, 'completeness' captures the essence of the property better?
>
> We agree. We have made the changes in the revised manuscript. We refer to it everywhere as "generality" now. We use generality as it is the standard term in the physics and chemistry literature.
>
> Q2. e3nn is not a chemical representation, it is a framework for training E(3)-equivariant neural networks (which may be used to learn representations of chemical systems).
>
> We agree. We eliminate the paragraph about e3nn from the manuscript.
>
> Q3. Section 3.1: "Since we believe the values to be missing at random, we utilize mean imputation, instead of discarding experimental data". This is a big assumption, do the results change significantly if missing values are removed?
>
> This is a great question! That was done only for certain columns of the photoswitches dataset because for some unknown reason the values are missing. We assumed the values were missing at random so we used mean imputation. We didn’t want to discard experimental data. We chose to include this dataset because it has been used before in recent literature see [1]. We don’t think the results would change significantly if missing values were removed, though we haven’t tested it.
>
> [1] GAUCHE: A Library for Gaussian Processes in Chemistry. Ryan-Rhys Griffiths, Leo Klarner, Henry
> Moss, Aditya Ravuri, Sang Truong, Yuanqi Du, Samuel Stanton, Gary Tom, Bojana Rankovic, Arian Ja-
> masb, Aryan Deshwal, Julius Schwartz, Austin Tripp, Gregory Kell, Simon Frieder, Anthony Bourached, Alex Chan, Jacob Moss, Chengzhi Guo, Johannes Peter D¨urholt, Saudamini Chaurasia, Ji Won Park,
> Felix Strieth-Kalthoff, Alpha Lee, Bingqing Cheng, Alan Aspuru-Guzik, Philippe Schwaller, Jian Tang.
> Conference on Neural Information Processing Systems. (2023)
>
> Q4. Please highlight your method in all tables for clarity.
>
> We make this suggested change. Our method in the tables is bolded.
>
> Q5. What exactly is the role of the random matrix denoting proton-electron interactions? Are these forces/energies learned?
>
> This is a great question! Yes it can either be learned or provided at initialization via a computation using force fields. This feature can then be used downstream for a task of interest. One may wish to learn the matrix however, we wish to provide that flexibility in the representation/model. In our experiments, however, we do not use this matrix at all (see the first bullet point for 4 in the other comment).
>
> Q6. Some of the definitions seem arbitrary. eg. where does the constant 2.8 fm come from in Definition 2.1?
>
> Good question! 2.8 fm is the ”classical” radius of an electron [2]. It is a bit simplistic and ignores quantum mechanics. We offset this simplicity by modeling an electron as a sphere paired with
> its corresponding wave function. Our representation is flexible enough that one can further add orbitals and diffuse functions after the fact to improve accuracy in this regard. The methods to do so, however, would increase the time complexity.
>
> [2] Haken, H.; Wolf, H.C.; Brewer, W.D. (2005). The Physics of Atoms and Quanta: Introduction to
> Experiments and Theory. Springer. p. 70. ISBN 978-3-540-67274-6.

---

### Official Review · Reviewer_VE9j · 2024-11-04

**Soundness:** 3
**Presentation:** 3
**Contribution:** 3
**Rating:** 6
**Confidence:** 2

**Summary:**

This paper presents a characterization method for polyatomic complexes that meets more properties compared to existing methods, such as Invariances, continuity and differentiability, and generality, generalizability.
It achieves strong results across multiple benchmarks.

**Strengths:**

1). Theoretical proof and experimental validation are sufficient.

**Weaknesses:**

1). The readability of the paper, including its organization and tables, could be improved. Could you provide more explanation for Figure 1?

2). Could you provide more explanations of the datasets and metrics used in the benchmarks section of the experiments? It might be a bit difficult for those without relevant background to understand.

**Questions:**

Refer to the questions in the Weaknesses.

---

> ### Author Response · Authors · 2024-11-26
> **Re: Official Review of Submission3311 by Reviewer VE9j**
>
> We would like to thank the reviewer for their insightful feedback. We address each weakness point by point.
>
> 1) The readability of the paper, including its organization and tables, could be improved. Could you provide more explanation for Figure 1?
>
> We agree that the readability of the paper could be improved. We make the following changes in the revised manuscript to improve readability:
> - We move the algorithm to the appendix.
> - We systematically reduce whitespace and extra spaces.
> - We introduce a new paragraph in the introduction clearly explaining the Graphical Abstract (Figure 1).
>
> 2) Could you provide more explanations of the datasets and metrics used in the benchmarks section of the experiments? It might be a bit difficult for those without relevant background to understand.
>
> We agree that the datasets should have more explanations and context. We address this by introducing a section overviewing the different datasets to better contextualize the tasks and results.
> - Please see section 3.1 of the revised manuscript for this change.

---

### Meta-Review · Area_Chair_HtX4 · 2024-12-20

**Metareview:**

**Summary**:
This paper contributes a novel representation for chemical systems. Specifically, the authors introduce 'polyatomic complexes' in order to characterize the structural and topological properties. Polyatomic complexes can account for symmetries (rotation, translation, and permutation invariances), continuity with respect to atomic positions, and topological accuracy upto electronic structure. The method can also accommodate arbitrary chemical compounds, providing well-defined representations. Experimental results are provided for multiple data: photoswitches, ESOL, FreeSolv and Matbench JDFT2D.

**Strengths**:
Reviewers acknowledged several strengths of this work including (a) the novelty of polyatomic complexes as an alternative to existing methods such as fingerprints and SMILES, (b) the flexibility to incorporate the physical characteristics of the target material, (c) comprehensive choice of datasets and baselines for empirical validation, and (d) the theoretical analysis.  The authors were also commended for being transparent about the experimental results being not particularly strong.

**Weaknesses**:
Reviewers also raised concerns about the readability of the paper (including lack of clarity in presentation, insufficient coverage of relevant chemistry background, details about baselines, datasets and metrics used for benchmarking), suboptimality of most experimental results compared to even simple baselines such as SMILES, and modeling inaccuracies from a chemical/physics perspective (such as treating electron as a single sphere).

The reviewers also pointed out some factual inaccuracies and made several suggestions for improvements, including, empirical assessments beyond property prediction such as evaluation of computational costs, generalizability, and ability to model long-range interactions.

The authors satisfactorily addressed some of these concerns during the response period, but many remained unresolved.

**Recommendation**:

The reviewers were almost unanimous in appreciating the novelty of the representation, and its theoretical guarantees. However, there is also consensus that experimental performance is not up to the mark and writing could be significantly improved. I agree with these concerns, and believe that this work requires major effort on these aspects. Therefore, I'm unable to recommend the paper positively in its current form.

**Additional Comments On Reviewer Discussion:**

All the relevant details already discussed above.

---

### Decision · Program_Chairs · 2025-01-22

Reject